# Exploring spatial feedbacks between adaptation policies and internal migration patterns due to sea-level rise

Lena Reimann [1,2,3] ✉, Bryan Jones[2], Nora Bieker[1], Claudia Wolff [1], Jeroen C.J.H. Aerts [3] & Athanasios T. Vafeidis [1]

Climate change-induced sea-level rise will lead to an increase in internal migration, whose intensity and spatial patterns will depend on the amount of sea-level rise; future socioeconomic development; and adaptation strategies pursued to reduce exposure and vulnerability to sea-level rise. To explore spatial feedbacks between these drivers, we combine sea-level rise projections, socioeconomic projections, and assumptions on adaptation policies in a spatially-explicit model ('CONCLUDE'). Using the Mediterranean region as a case study, we find up to 20 million sea-level rise-related internal migrants by 2100 if no adaptation policies are implemented, with approximately three times higher migration in southern and eastern Mediterranean countries compared to northern Mediterranean countries. We show that adaptation policies can reduce the number of internal migrants by a factor of 1.4 to 9, depending on the type of strategies pursued; the implementation of hard protection measures may even lead to migration towards protected coastlines. Overall, spatial migration patterns are robust across all scenarios, with out-migration from a narrow coastal strip and in-migration widely spread across urban settings. However, the type of migration (e.g. proactive/reactive, managed/autonomous) depends on future socioeconomic developments that drive adaptive capacity, calling for decision-making that goes well beyond coastal issues.

There is consensus in the scientific literature that the impacts of climate change will affect migration patterns globally, likely leading to an increase in the number of migrants[1–6]. Climate change will not only affect environmental drivers of migration, for example, through an increase in extreme weather events such as droughts, heat waves, or floods[7], but all migration drivers (i.e., economic, political, social, and demographic), for instance through crop losses or the adoption of climate change policy[3,8,9]. While climate change is also expected to lead to higher international migration flows[2,10], the majority of migrants are more likely to move within country borders ("internal migration") and

over short distances[11,12]. Currently, most internal migration takes place from rural to urban areas; these patterns are expected to be reinforced with progressing climate change[13,14]. Recent World Bank reports project up to 143 million climate change-related internal migrants in Sub-Sahara Africa, South Asia, and Latin America by 2050[15], and up to 216 million when additionally accounting for North Africa, Eastern Europe and Central Asia, and East Asia and the Pacific[16].

Sea-level rise (SLR)-related migration has received increasing attention in recent years, particularly as the impacts of SLR, such as submergence of low-lying land, saltwater intrusion, increasing coastal

[1]Coastal Risks and Sea-level Rise Research Group, Department of Geography, Kiel University, Ludewig-Meyn-Straße 8, 24118 Kiel, Germany. [2]CUNY Institute for Demographic Research (CIDR), City University of New York, 135 E 22nd St, New York City, NY 10010, USA. [3]Institute for Environmental Studies (IVM), Vrije Universiteit Amsterdam, De Boelelaan 1111, 1081 HV Amsterdam, The Netherlands. ✉e-mail: lena.reimann@vu.nl

 1

erosion as well as more frequent and intense coastal flooding due to extreme sea levels (ESL) may threaten the livelihoods of entire islands or nations[17–19]. Also, beyond small island states, SLR will affect internal migration flows considerably, mainly driven by highly urbanized coastal areas[20–23]. In 2010, about 27% of the global population and 34% of the urban population lived in a coastal strip that covered 9% of the global land area[21]. These spatial patterns are expected to continue in the future due to progressing urbanization as well as the continued high attractiveness of coastal areas for human settlement[24–27]. Therefore, SLR may result in two types of internal migration responses: in permanent migration due to slow-onset impacts such as submergence of land or coastal erosion, and in temporary displacements during coastal flooding due to ESL[18,28].

The current literature assessing SLR-related internal migration primarily assumes that the population exposed to SLR will (be forced to) migrate autonomously[15,16,29–32]; this notion is often referred to as "migration as adaptation"[13]. Studies that assess how different adaptation strategies may influence SLR-related internal migration at supranational (i.e., continental to global) scales[33] are scarce. A recent global-scale study[34] found that submergence due to SLR could result in ~35 million autonomous migrants until 2100, assuming a mean SLR of 1.1 m and cost-effective hard protection at 3.4% of the coastline. Including managed retreat out of the 1-in-10-year coastal floodplain in unprotected stretches of the coast, this number could increase to about 40 million migrants by 2100. Thus far, such assessments do not explore the potential effects of adaptation policy scenarios that integrate protection, accommodation, and managed retreat strategies on SLR-induced internal migration[18], a research need that has been raised in recent years[35,36].

Furthermore, few studies at the supra-national scale analyze future spatial patterns of climate change-related internal migration (exceptions are refs. 15,16,37) as previous work has focused on estimating the total number of migrants. Potential migrant destinations are more commonly investigated in national- to local-scale assessments where observed data of internal migration flows are more readily available, e.g., from national surveys or tax data[38–42]. At the same time, assessing plausible spatial patterns of SLR-related internal migration can help identify migration hotspots in sending as well as receiving areas that can support decision-making in anticipating and managing internal migration flows[11,36,43]. Accounting for a range of coastal adaptation strategies in such assessments can provide important insights into the spatial feedback between adaptation and spatial patterns of migration flows, for instance, related to the so-called safe development paradox ("levee effect"), where protected areas become more attractive for human settlement, thus resulting in an increase in exposure[44–47].

We address the above research gaps by analyzing plausible internal migration patterns due to SLR, specifically focusing on exploring the effects of coastal adaptation policies—including protection, accommodation, and managed retreat strategies—on the number, timing, and spatial distribution of internal migrants. Our modeling approach builds on a gravity-based population downscaling model designed to produce spatial population projections[48,49], also used in recent reports on internal migration due to climate change[15,16,37,50]. Reimann et al.[51] extended the model to account for inland-coastal migration in addition to rural-urban migration (called "CONCLUDE"). They calibrated and validated CONCLUDE to the Mediterranean region, characterized by high population densities and urbanization levels in the immediate coastal zone[52]. They further differentiated two geographical regions, the northern versus the southern and eastern Mediterranean (Supplementary Table 1), to account for the differences in socioeconomic development and adaptive capacity across the region[53,54].

In this study, we extend CONCLUDE in order to account for plausible spatial feedbacks between SLR, adaptation policies, and spatial patterns of internal migration flows in the Mediterranean until 2100, using exploratory modeling based on what-if explorations of the future[55,56]. For this purpose, we employ the current scenario framework in climate change research[57,58], integrating SLR projections based on the representative concentration pathways (RCPs)[59], population and urbanization projections based on the shared socioeconomic pathways (SSPs)[60], and shared policy assumptions (SPAs)[61] for coastal adaptation that we have developed specifically for this study (Supplementary Text 1). Following common practice in supra-national scenario-based modeling studies[15,16,34,62–67], we use a selected set of plausible integrated scenarios that span the relevant uncertainty range regarding future climatic and socioeconomic conditions.

Our adaptation policy scenarios (i.e., SPAs) include a range of coastal protection, managed retreat, and accommodation strategies that differ across the integrated scenarios (Fig. 1). "Build with Nature" is a sustainable scenario (SSP1) with low SLR (RCP2.6) that combines a range of hard protection, managed retreat, and accommodation strategies; "Save Yourself" is characterized by regional rivalry (SSP3), with moderate SLR (RCP4.5) and hard protection limited to densely populated locations; "Hold the Line" involves high SLR (RCP8.5) due to the dependence on fossil fuels (SSP5) and primarily relies on large-scale implementation of hard protection measures (see Methods "Integrated scenarios" and Supplementary Table 2 for additional detail). Rather than predictions of future migration, our results aim to understand plausible trends in potential intensity and spatial patterns of internal migration driven by SLR and SLR-related adaptation policies from 2020 to 2100 due to a range of uncertainties inherent in the study (see Discussion section).

## Results

### The effects of adaptation on total migrant numbers

Across the Mediterranean region, submergence due to SLR may result in over 20 million internal migrants by 2100 if no adaptation policies are pursued (Fig. 2a, solid lines). This total number is largely independent of the SSP-RCP combination assessed; however, in SSP1-RCP2.6 the highest increase in migrant numbers takes place towards the end of the century, while in the other two scenarios, this increase occurs earlier. With adaptation policies (dashed lines), the cumulative number of migrants until 2100 decreases by a factor of 1.7 and 1.4 under the "Build with Nature" (SSP1-RCP2.6) and "Save Yourself" (SSP3-RCP4.5) scenarios, respectively, and by a factor of 9 under "Hold the Line" (SSP5-RCP8.5), where large-scale hard protection is pursued (see Methods "Accounting for the effects of adaptation on internal migration"), thereby resulting in the lowest number of migrants (~2 million) by 2100 (red dashed line). In "Build with Nature" (blue dashed line), migration is at a higher level from the beginning (roughly 3 million in 2020) and increases gradually to about twelve million migrants until 2100 due to the proactive implementation of accommodation and managed retreat strategies. In "Save Yourself" (yellow dashed line), the number of migrants increases nearly tenfold to ~15 million (2100), mostly driven by autonomous migration as sea levels rise.

Similar patterns can be observed when considering the number of migrants in the course of the century (Fig. 2b). Without adaptation policies (left panel), migration is relatively high and gradually levels off in the first half of the century across all SSP-RCP combinations. In the second half of the century, a tipping point is reached where around five million people migrate autonomously, which occurs earlier under SSP3-RCP4.5 and SSP5-RCP8.5 compared to SSP1-RCP2.6, driven by an earlier acceleration in SLR under the former two scenario combinations[68]. This tipping point is predominantly driven by large-scale submergence in the Nile delta, with 82% (SSP5-RCP8.5) to 90% (SSP1-RCP2.6) of all internal migrants projected for these time steps living in Egypt (Supplementary Data 1). With adaptation policies (right panel), proactive adaptation in "Build with Nature" leads to the largest share of the total migration taking place in the first half of the century

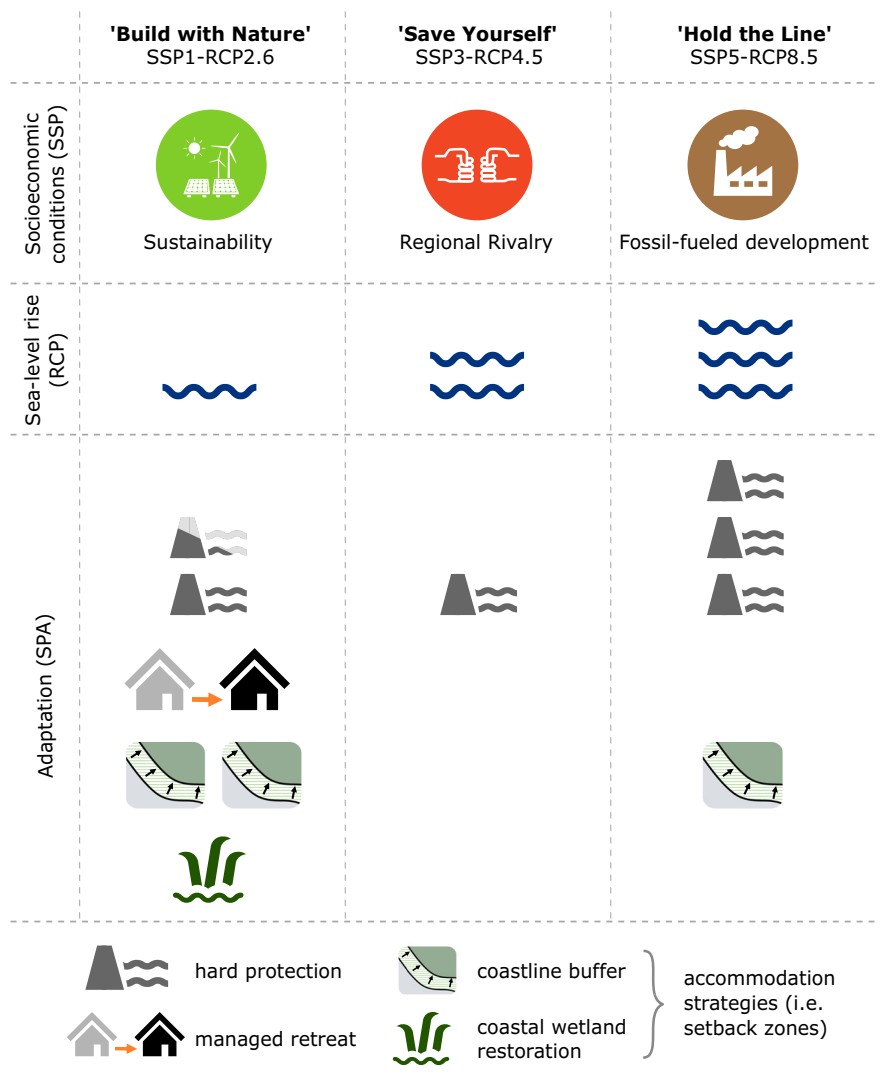

**Fig. 1 | Integrated scenario assumptions used in this study.** The number of symbols represents intensity. Columns show assumptions per integrated scenario; rows allow for a comparison across scenarios. SSP shared socioeconomic pathway, RCP representative concentration pathway. Please see Methods "Integrated scenarios", Supplementary Text 1 and Supplementary Table 2 for further context. The SSP icons were reused from ref. 146: Extending the shared socioeconomic pathways (SSPs) to support local adaptation planning—A climate service for Flensburg, Germany. Futures 127, 102691 (2021)—CC BY 4.0.

and gradually declining until 2100, while in "Save Yourself," a migration tipping point is reached at the same time as without adaptation policies; however, the number of migrants is reduced by approximately one third due to the implemented protection measures. Under the "Hold the Line" scenario, migration flows are at a low level, with the highest number of migrants in the first half of the century. Under all scenario combinations, migration flows by 2100 are dominated by urban migrants (solid bars): without adaptation policies, 70–86% of all migrants are projected to be urban, while adaptation policies may result in a lower share of urban migrants (56–68%) owing to the assumption that most protection strategies are implemented in urban settings (see Methods "Accounting for the effects of adaptation on internal migration").

Comparing the cumulative number of migrants across geographical regions by 2100 (Fig. 3), we see consistently higher total migration in southern and eastern Mediterranean countries under all scenarios except the "Hold the Line" (SSP5-RCP8.5) scenario, where more people migrate in the North (-1.3 million) compared to the South and East (roughly 1 million). Without adaptation policies, total migration until 2100 is approximately three times higher in the South and East (about 14-15 million) than in the North (-5–6 million) under all

scenario combinations. With adaptation policies, the South and East sees roughly three million more migrants than the North in "Build with Nature" (7.5 versus 4.5 million), and almost three times as many in "Save Yourself" (11 versus 4 million), driven by the assumption that fewer adaptation measures are implemented in southern and eastern countries due to their lower adaptive capacity (see Methods "Accounting for the effects of adaptation on internal migration"). Despite the considerable differences in total migrant numbers, adaptation policies have a similar effect on reducing the number of migrants in both geographical regions: in the northern Mediterranean, the implementation of adaptation policies is projected to reduce migration flows by 23% ("Build with Nature") to 79% ("Hold the Line"), and by 29% ("Save Yourself") to 94% ("Hold the Line") in the South and East.

## The effects of adaptation on spatial migration patterns
If no adaptation policies are pursued (Fig. 4a), most internal migration takes place out of a narrow coastal strip submerged by SLR into inland locations, in particular into urban settings. The migration intensity depends on the amount of SLR (see Methods "Modeling SLR-induced internal migration") as well as on population growth and urbanization

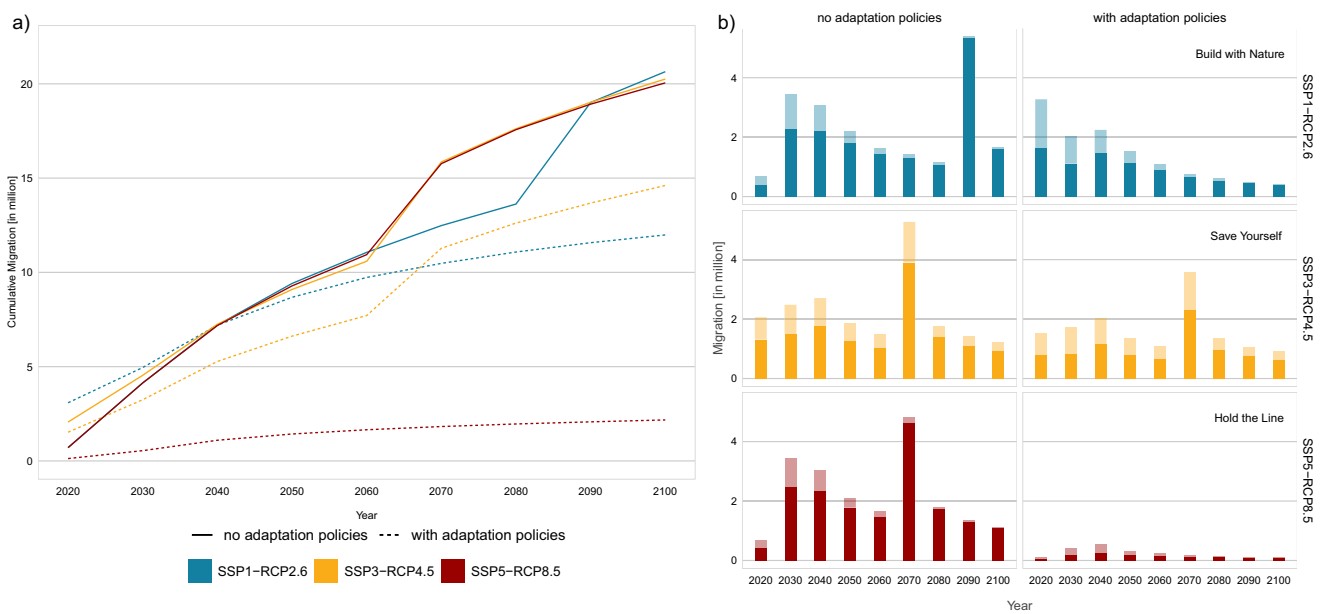

**Fig. 2 | Internal migration due to sea-level rise without and with adaptation policies. a** Cumulative migration summed up across the century; **b** Number of migrants per decade in rural versus urban settings without adaptation policies (left panel) and with adaptation policies (right panel). SSP shared socioeconomic pathway, RCP representative concentration pathway. Note that the numbers presented here demonstrate indicative trends of potential internal migration and give an indication of the timing of migration in the course of the 21st century, with a focus on the difference between the implementation of adaptation policies compared to no adaptation policies.

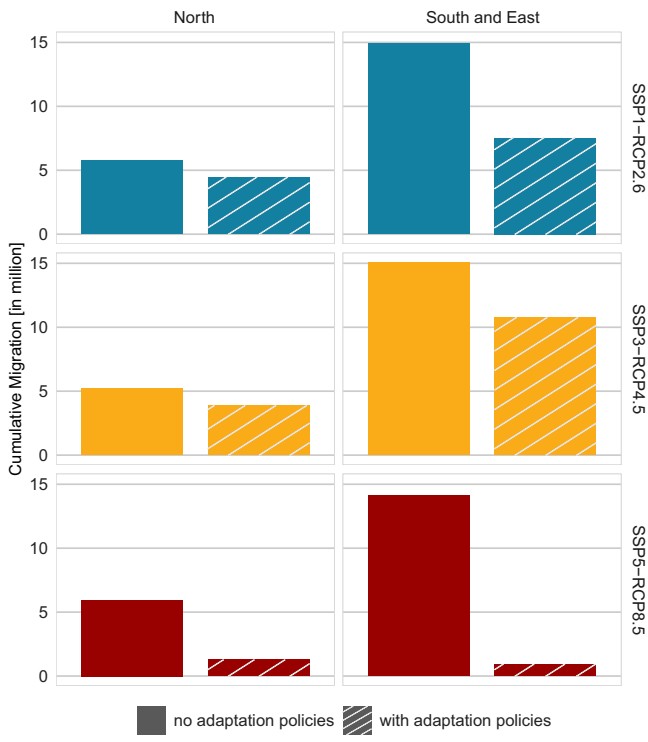

**Fig. 3 | Cumulative internal migration by 2100 in the two geographical regions without and with adaptation policies.** SSP shared socioeconomic pathway, RCP representative concentration pathway.

rates per scenario combination (see Supplementary Fig. 1 for the baseline "no SLR" population projections). Moreover, these spatial patterns are reinforced until 2100 as the attractiveness of coastal locations decreases with decreasing population and inland locations become more attractive as population increases. With adaptation policies (Fig. 4b), the overall spatial migration patterns remain the same, but are less pronounced, in particular under the "Hold the Line" (SSP5-RCP8.5) scenario where large-scale hard protection reduces migration considerably.

Based on the difference between the spatial migration patterns under the "with adaptation policies" scenarios (Fig. 4b) and the "no adaptation policies" reference projections (Fig. 4a), we illustrate the effects of adaptation policies on spatial migration patterns (Fig. 4c): under all policy scenarios, adaptation policies largely reverse the migration patterns caused by SLR: population grows substantially in protected stretches of the coast, resulting in high population concentrations in these locations (green colors). This levee effect is particularly prominent in southern and eastern Mediterranean countries under "Save Yourself" (Fig. 4c, panel 2) and in northern countries under "Hold the Line" due to high population growth under these scenarios in the respective regions (see Supplementary Fig. 4). Migration to protected coastlines largely occurs out of urban areas (Fig. 4c, all panels; pink colors), both from locations in close proximity to the coast as well as from larger cities located inland such as Damascus, Jerusalem, and Cairo. In "Build with Nature" (Fig. 4c, panel 1), spatial migration patterns differ from the "no adaptation policies" reference projections to a limited degree: adaptation policies primarily result in a further decline in population in coastal locations not protected by hard protection measures due to the implementation of setback zones (i.e., coastline buffer plus coastal wetland restoration) and managed retreat in frequently flooded locations. Although less pronounced than in the other two policy scenarios, the levee effect can be observed in protected stretches of the coast as well.

## Discussion

This study explores future internal migration due to SLR in the Mediterranean region, based on a set of integrated RCP, SSP, and SPA scenarios. We focus on understanding the intensity as well as the spatial and temporal patterns of permanent migration driven by SLR-related submergence and adaptation policies. These migration processes can take place over short distances (i.e., to the next raster cell)

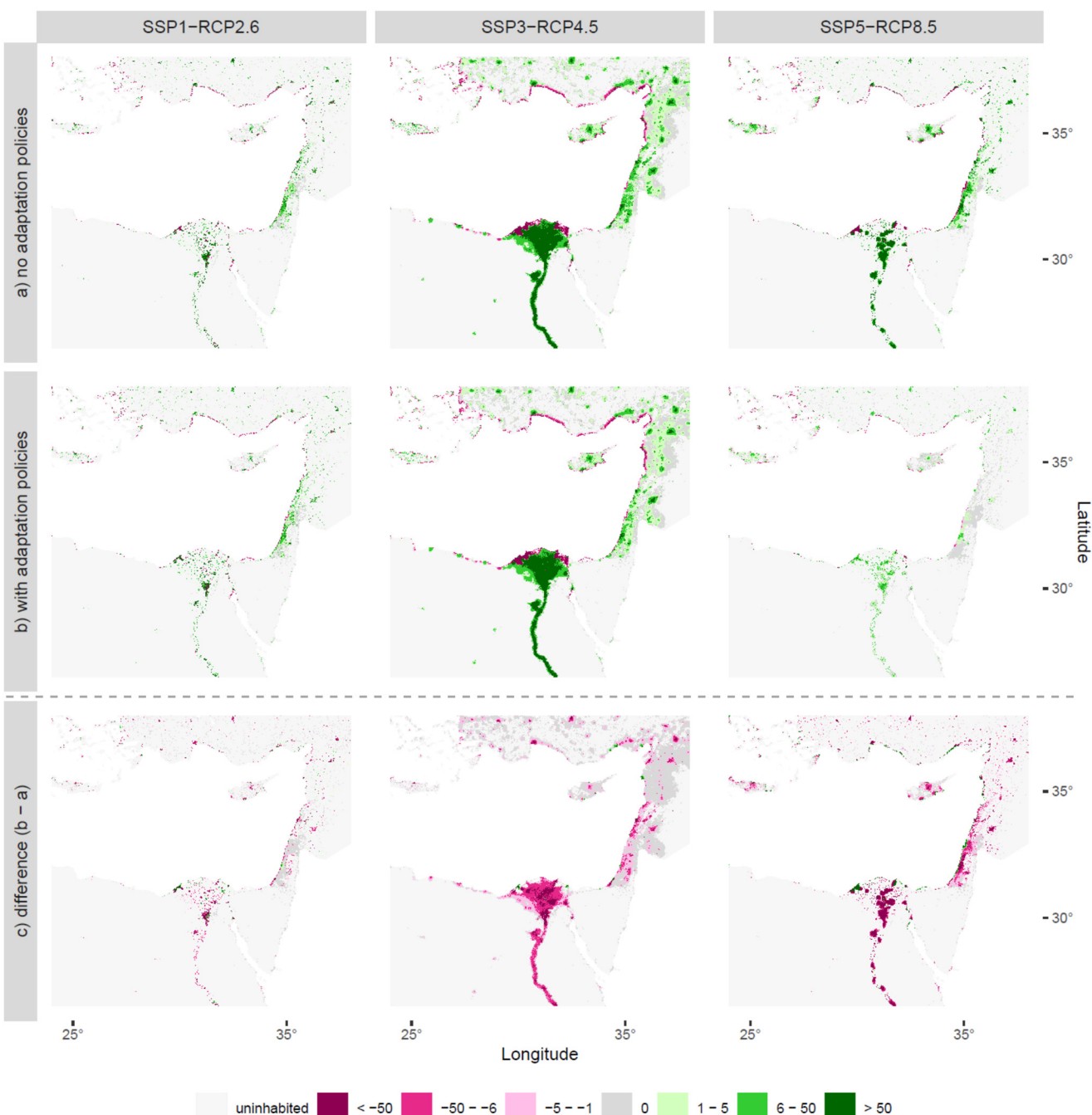

**Fig. 4 | Spatial migration patterns due to sea-level rise in the eastern Mediterranean in 2100 by SSP-RCP combination. a** Migration without adaptation policies (upper row); **b** Migration with adaptation policies (middle row); both **a** and **b** are compared to the "no sea-level rise" baseline projections. **c** Shows the difference between **a** and **b** (lower row), thereby presenting the effect of adaptation policies on internal migration patterns compared to the "no adaptation policies" reference projections. Pink colors show out-migration; green colors show in-migration. SSP shared socioeconomic pathway, RCP representative concentration pathway. Supplementary Figs. 2–4 show the entire Mediterranean of rows **a–c**, respectively.

and the same people may be forced to migrate multiple times in the course of the century (see Methods), thereby resulting in high numbers of projected internal migrants until 2100. Our results show that—without the implementation of adaptation policies—the total number of migrants until 2100 is largely independent of the SSP-RCP scenario combination despite different amounts of projected SLR. This effect is driven by the underlying population[69] and urbanization projections[70] as well as urban sprawl assumptions[51]: in SSP1, urban areas are very attractive (i.e., they have a high population potential) due to high urbanization rates and limited urban sprawl. As urban settlements are concentrated along the Mediterranean coastline, a large share of

migrants is distributed in the immediate coastal zone, which is affected by submergence due to SLR in the next time step(s), thus leading to high migrant numbers until 2100. This effect is less pronounced under SSP5, where urban sprawl is high and under SSP3, where urbanization rates are low. These patterns demonstrate that socioeconomic developments, as described in the SSPs, have a larger influence on the number of migrants compared to the amount of SLR, thereby stressing the need for appropriate adaptation policies to manage future coastal risks.

We find that adaptation policies can considerably influence the number and spatial patterns of future migration. While adaptation

reduces the potential number of internal migrants under all scenario combinations (Figs. 2, 3), this outcome may not be desirable from a risk management perspective due to the levee effect[44,47]: it results in a spatial feedback loop that gradually increases the number of people living in stretches of the coastline protected by hard protection measures until 2100 (Fig. 4c). This safe development paradox is particularly pronounced under the "Hold the Line" scenario where more than one-quarter of the coastline is protected from SLR (see Methods "Accounting for the effects of adaptation on internal migration"), thereby resulting in population growth and hence high residual risk, which can lead to high impacts in case of protection failure during ESL events. Although this scenario is characterized by low adaptation challenges (i.e., high adaptive capacity), resulting in well-managed proactive adaptation, it leads to a path dependence where continuous upgrading of protection measures is required as sea levels rise. With SLR projected to accelerate from 2050 and well beyond 2100 under RCP8.5[68], a tipping point may be reached where hard protection measures are no longer feasible, therefore requiring a new policy action (so-called "adaptation tipping point")[71,72], or causing large-scale out-migration[18].

In "Build with Nature", also characterized by low adaptation challenges, hard protection measures are restricted to densely populated urban areas and are complemented with accommodation strategies (i.e., setback zones) and managed retreat. Although migrant numbers are high, migration is proactive and well-managed, which leads to comparatively low residual risk, also beyond 2100, due to relatively low SLR in RCP2.6. High adaptation challenges in "Save Yourself" result in limited protection of about 3% of the coastline, and people migrate out of high-risk locations once the impacts of SLR are felt, which will potentially cause high damages in submerged locations. This reactive form of migration is particularly challenging in less developed countries as a large share of the population may not be able to migrate autonomously ("trapped populations")[13,73], which would potentially be the case in the Mediterranean South and East under this scenario[60]. It is therefore important to consider future socioeconomic developments in addition to the total number of migrants when devising migration-related management strategies.

To establish the robustness[56] of our results across the six integrated scenarios (i.e., SSP1-RCP2.6; SSP3-RCP4.5; SSP5-RCP8.5, each without and with adaptation policies), we have calculated migration hotspots based on the upper/lower 10% of in- and out-migration per scenario (see Methods "Calculation of migration hotspots"). This approach enables us to establish locations where scenarios consistently project high levels of internal migration with moderate to high confidence (Fig. 5). We find migration hotspots to be similar across all scenarios—independent from the implementation of adaptation policies—with high out-migration from a narrow coastal strip that leads to widespread in-migration across urban settings, particularly across large urban centers. While adaptation policies may alter some of these hotspots (most notably in the Nile Delta, see Supplementary Fig. 5), many regions see high levels of in- and out-migration even if adaptation policies are pursued. Similar to previous work[15,16], the hotspot analysis shows that our results are robust across SLR, socioeconomic, and adaptation policy scenarios. It can therefore serve as a basis for establishing priority areas for policy planning regarding coastal adaptation and the management of migration flows.

Our results are subject to a range of uncertainties that need to be considered. These uncertainties stem from the underlying scenario assumptions; the modeling approach; and the input data used. As such, the results represent plausible trends in internal migrant numbers and spatial migration patterns until 2100, rather than predictions of total migrant numbers. We discuss examples of uncertainties relevant to this work in the following sections and refer to refs. 23,74 for comprehensive analyses and discussion of uncertainties in supranational coastal risk assessments.

As we explore a selected set of integrated scenarios, we may not cover the full range of uncertainty regarding future socioeconomic and climatic conditions in driving internal migration. By accounting for SSPs with both low and high challenges for adaptation as well as the respective plausible SLR scenarios based on the RCPs, we are nevertheless confident that we span the relevant uncertainty range related to adaptation policies (see Methods "Integrated scenarios"). Furthermore, in the adaptation policy SPAs, we assume adaptation policies to be effective immediately (i.e., in 2020) until the end of the century. However, the implementation of adaptation policies has long lead times[75,76] and hard protection strategies need regular maintenance to facilitate their protective function[62,77]. Similarly, as official guidelines for designating managed retreat zones currently do not exist for the Mediterranean region, we rely on expert judgment to determine these zones based on ESL return periods. We assume that all unprotected coastal stretches become such zones, although so far, managed retreat has been implemented in specific local cases only[18,78], but might become a more widespread solution as SLR accelerates[72].

Additionally, the gravity-based migration modeling approach may introduce uncertainties that need to be considered. Gravity models are designed to reflect aggregate human behavior rather than individual decisions, favoring densely populated locations in close proximity over less densely populated and more distant locations[79,80]. While this underlying assumption may be largely applicable when modeling slow demographic change in a spatially explicit manner, it may not hold true when it comes to the impacts of SLR, which may result in new migration patterns such as temporary displacements during coastal flooding due to ESL, followed by limited return migration as observed after Hurricane Katrina[81] or potential abandonment of coastal areas due to a domino effect of successive out-migration from the coast[82,83]. Further, it remains to be seen whether coastal cities protected by hard protection measures continue being attractive for human settlement with increasing SLR. If people migrate inland, another open question is the distance that people are willing to migrate, particularly as the current literature suggests a preference for short-distance migration[11,12]. However, the current model setup disregards the migration distance from sending (coastal) to receiving (inland) regions. Understanding of these processes is still limited due to a lack of empirical evidence regarding the impacts of SLR on adaptation and migration decisions[18,35,84], and potential socioeconomic tipping points[82,85,86]. This lack of empirical data also hampers the validation of the model. While data on observed internal migration flows are available at the administrative unit level for selected Mediterranean countries, e.g., through the IPUMS (Integrated Public Use Microdata Series) database (five countries)[87] or the IMAGE (Internal Migration Around the GlobE) project (eight countries)[88,89], these flows do not necessarily reflect SLR (policy)-related migration as the observed impact of SLR on internal migration is still limited; and adaptation policies like those explored in this study currently do not exist.

The global data used as model input introduce further uncertainties. The decadal SLR, population, and urbanization projections employed are subject to uncertainties in the underlying assumptions and models used for producing these projections. Further, we model the coastal floodplain with the help of digital elevation data that are based on a digital surface model[90], meaning that the floodplain may be underestimated in built-up locations[91,92]. The satellite-based GHSL population and settlement rasters used to characterize the baseline urban versus rural population distributions introduce additional uncertainties due to (a) the approach used for spatially distributing the population[23,93]; (b) overconcentration of the population as not all built-up land is detected in satellite imagery[94]; and (c) inconsistencies in the definition of urban areas[23,95]. Additionally, as region-wide data on current coastal protection levels are not available, we have to assume that all land that is currently located below mean sea level will be submerged in the first modeling step (i.e., 2020), which is one reason

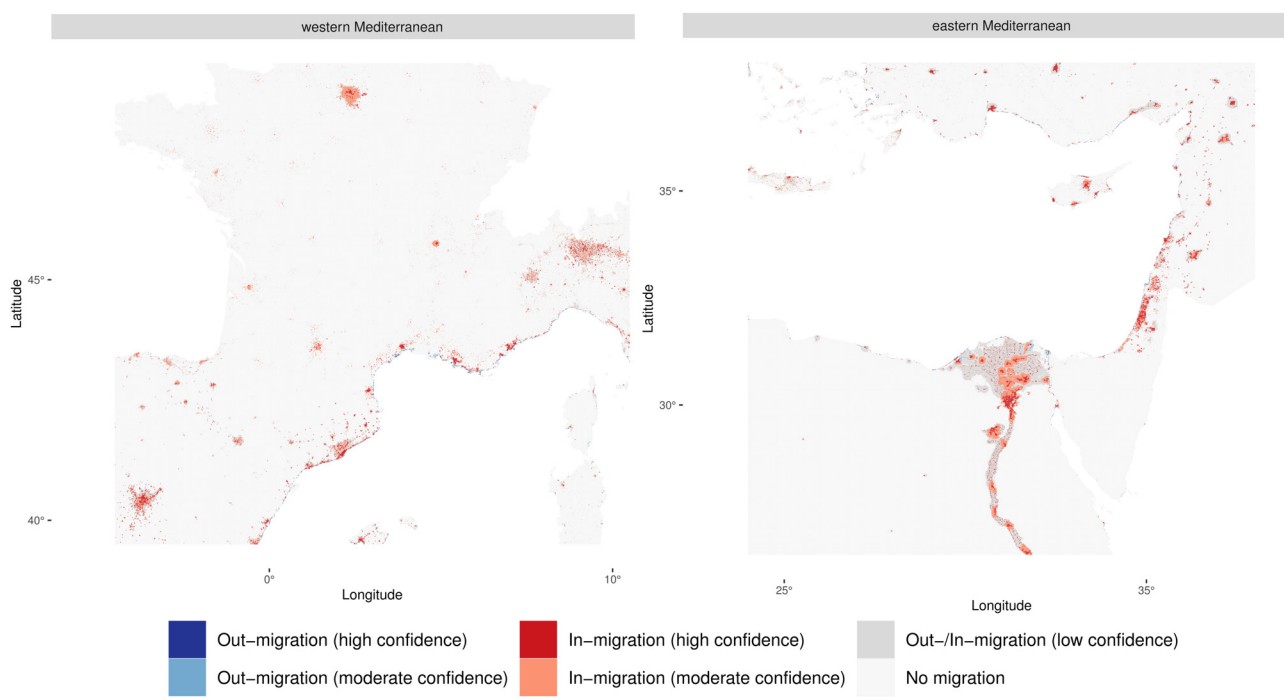

**Fig. 5 | Migration hotspots of out- and in-migration across all scenarios (i.e., no adaptation policies and with adaptation policies) in the western versus eastern Mediterranean by 2100.** The hotspots are calculated based on ref. 16 (see Methods "Calculation of migration hotspots"). Supplementary Fig. 5 shows the entire Mediterranean.

for the high migrant numbers projected at the beginning of the century under the 'no adaptation policies' scenarios, as suggested in recent work[96].

In order to better understand the effects of different input datasets, modeling approaches, and scenario assumptions on the projected number and spatial patterns of internal migration, a systematic sensitivity analysis that quantifies the uncertainties caused by these effects is needed. However, such an analysis was beyond the scope of this study. Despite the uncertainties discussed here, we are confident that our study provides a valid initial estimate of the intensity and spatial patterns of future SLR-related migration by exploring a range of adaptation policy scenarios.

While the aim of this study was to explore plausible spatial feedbacks between SLR, adaptation policies, and internal migration, it is challenging to draw specific recommendations for policymaking from it due to the uncertainties discussed above. For instance, adaptation policies vary across and within countries due to heterogeneity in policy making[97,98], which is driven by the overall adaptive capacity per country[99], and varies considerably across the Mediterranean due to differences in socioeconomic development[53]. Thus, future work can refine the adaptation policy scenarios (i.e., SPAs) developed here, accounting for national- to local-level conditions in driving adaptation decision-making. Furthermore, migration decisions depend on individual characteristics that determine a person's ability and willingness to migrate, such as demographics, socioeconomic status, and ethnicity[3,18]. So far, these characteristics of social vulnerability have hardly been accounted for at this scale of analysis (exceptions are refs. 16,37), although including vulnerability allows for modeling more diverse migration patterns, especially with regard to trapped populations[36,84].

To account for migration decisions at the individual level, top-down modeling approaches like CONCLUDE can be combined with bottom-up approaches such as agent-based models (ABMs)[36,100]. According to ref. 36, top-down approaches are suitable for establishing hotspots of migration, but tend to oversimplify the migration response. Therefore, a possible way forward is using the results of top-down approaches as boundary conditions for agent-based approaches, and accounting for more refined migration behavior based on agent decisions and individual preferences[35,101]. In this manner, dynamic feedbacks between SLR and adaptation uptake can be explored, allowing for the analysis of emergent agent behavior with progressing SLR[102,103]. To explore the effect of adaptation policies on individual adaptation decisions (including autonomous migration), decisions of other agents such as governments can also be included[104]. While ABMs have been primarily applied at local to national scales (e.g., refs. 42,100,105) due to the high computational demands and data requirements[104], a recent study has applied an ABM at supra-national scale to assess the future risks of river flooding in Europe[47].

Combining modeling approaches would also contribute to harmonizing migration modeling studies as results of recent supra-national work (refs. 15,16,34,37) are difficult to compare, also due to differences in the input data used[23,74,106]. Transparent reporting of scenario assumptions, modeling approaches, and input data is important to contextualize uncertainties in the modeling chain[27,74]. Quantifying these uncertainties systematically in a sensitivity analysis, by comparing migration modeling approaches or the effect of different input data (e.g., population, urban settlements, SLR projections, elevation) on model results is urgently needed in follow-up research.

Future work can further explore new ways to calibrate and validate models at a supra-national scale with the help of observed migration flows. As CONCLUDE was calibrated and validated using spatial population data of different time steps (i.e., GHS-POP[107]), observed migration patterns were only captured indirectly[51]. In data-scarce regions, mobile phone data can provide additional insights into internal migration flows[108,109]. Similarly, empirical surveys can help establish drivers of SLR-related migration, thereby providing a basis for calibrating and validating migration models based on observed as well as anticipated migration and adaptation behavior due to SLR[84] (as recently done in ref. 100). Furthermore, it is worth exploring the individual effect of each adaptation measure (i.e., hard protection; coastline buffer; restoration of coastal wetlands; managed retreat) on the number and spatial patterns of migration, which we did not assess

in this study as our aim was to explore three integrated adaptation policy scenarios of combined adaptation measures (Fig. 1). Accordingly, to span the full range of uncertainty in future socioeconomic and climatic conditions, all plausible scenario combinations as established in previous work[58,110] can be explored. Last, while we have focused on SLR-induced migration, other climate change impacts such as droughts and extreme heat will additionally affect migration flows[3], which can be integrated into CONCLUDE as well[15,16,37].

Finally, this study constitutes an important step forward in modeling plausible feedbacks between SLR, adaptation policies, and internal migration in a spatially explicit manner, which allows for analyzing sending versus receiving regions of migrants in the course of the 21st century. The RCP-SSP-SPA scenario framework provides a flexible tool for exploring these spatial feedbacks. Importantly, we must stress the need to go beyond the assessment of migrant numbers, but to contextualize these with the socioeconomic developments in each scenario that drive adaptation policies: a high number of migrants can result from the implementation of proactive strategies, while a low number of migrants can lead to high residual risk due to the levee effect. Decision-making related to adaptation planning and the management of internal migration flows needs to consider these spatial feedbacks in order to avoid maladaptation and to facilitate sustainable development in coastal areas and well beyond.

## Methods
### Integrated scenarios
We selected a set of three integrated scenario combinations, ensuring that we covered the uncertainty range with regard to the future amount of SLR (i.e., RCPs), society's adaptive capacity determined by socioeconomic conditions (i.e., SSPs), and the type of adaptation strategies pursued (i.e., SPAs). Selecting a set of plausible scenario combinations is common practice in supra-national scenario-based modeling studies[15,16,34,62–67]. We based our integrated scenario selection on the most plausible SSP-RCP combinations as established in previous work[58,110].

As a first step, we selected the SSPs to be investigated in this study. From the five SSPs that were developed based on their challenges for climate change mitigation and adaptation[111], we used two with low challenges for adaptation (i.e., SSP1, SSP5) and one with high challenges for adaptation (i.e., SSP3). We selected SSP1 and SSP5 as socioeconomic developments in both pathways differ: SSP1 focuses on sustainable development, while SSP5 is driven by fossil-fueled development, both leading to low adaptation challenges. In SSP3, adaptation challenges are high due to regional rivalry associated with limited socioeconomic development[60]. Next, we combined the selected SSPs with plausible sea-level rise scenarios based on the RCPs, ensuring that we covered the uncertainty range of future SLR. Out of the set of plausible combinations[58,110], we used the following three: SSP1-RCP2.6, SSP3-RCP4.5, and SSP5-RCP8.5. These constituted our "no adaptation policies" reference scenarios.

Furthermore, we developed SPAs for coastal adaptation based on the characteristics of each SSP, as the challenges for adaptation described in each SSP define the type of plausible adaptation policies[61,112]. We established the current state-of-the-art regarding adaptation practices as a starting point for our assumptions by reviewing the existing literature regarding coastal adaptation in general[18,19,113–117] and specifically in the Mediterranean region[118–123]. In order to explore adaptation policies with the largest differences in plausible spatial migration outcomes, we developed three distinct sets of coastal adaptation SPAs: "Build with Nature" (SSP1-RCP2.6), "Save Yourself" (SSP3-RCP4.5), and "Hold the Line" (SSP5-RCP8.5). The narrative description of each coastal adaptation SPA can be found in Supplementary Text 1; Supplementary Table 2 provides an overview of the developed coastal adaptation assumptions (see "Accounting for the effects of adaptation on internal migration" below for the

quantification of each scenario combination). We refrained from analyzing individual adaptation measures separately as we did not consider them to be plausible; for instance, solely relying on managed retreat would be implausible as large shares of the coastal population would have to migrate, in particular out of urban locations that would likely receive some degree of hard protection[34,77]; the same applied to the implementation of setback zones[123].

### Modeling approach
We used the gravity-based population downscaling model CONCLUDE that was designed to produce raster-based population projections at a spatial resolution of 30 arc seconds (roughly 1 km at the equator) and a temporal resolution of 10-year time steps, accounting for inland-coastal migration in addition to rural-urban migration and spatial development patterns (i.e., urban sprawl)[51]. CONCLUDE is an extended version of INCLUDE[48,49,124] which has been applied in a wide range of applications, for instance, related to heat extremes[63,125,126], vector-borne diseases[64], as well as internal migration due to climate change[15,16,37]. The model is based on Newton's law of gravity and uses the notion of "population potential"[127], where densely populated locations are attractive for human settlement due to e.g. job opportunities and income differentials[80]; and attractiveness decreases with increasing distance ("distance decay") owing to factors such as transport costs and travel times[79].

CONCLUDE (i.e., *CO*astal *INCLUDE*) iterates through each time step $t$ to calculate a population potential per grid cell ($v_i$) based on the spatial population distribution (Eq. 1). The potential is calculated by combining the distance-decay effect with local characteristics that drive the attractiveness of specific locations, and is weighted with a spatial mask that masks out all land not available for human settlement, e.g., due to the presence of water, steep slopes, or deserts:

$$v_i(t) = l_i \left( \sum_{j \in N_i} P_j(t) e^{-\beta d_{ij}} + A_i P_i(t) \right). \quad (1)$$

where $l_i$ is the spatial mask, $P$ is the population of cell $j$ or $i$, $\beta$ is a parameter that reflects the strength of the distance-decay effect, and $d_{ij}$ is the distance between cells $i$ and $j$, determined by the gravity window within which the distance-decay effect applies, which also defines the number of neighboring cell indices $N_i$. The local attractiveness of cell $i$ is reflected in the factor $A_i$, which has been established during model calibration and is kept constant until 2100. Based on the population potential of each cell, CONCLUDE distributes national-level population projections spatially, additionally differentiating urban versus rural populations in coastal versus inland locations[51].

For this study, we used the model calibrated and validated for the Mediterranean region, as described in ref. 51, along with the spatial population projections produced with it[128]. These projections used the GHS-POP population data at 30 arc seconds resolution (2019 version)[107] as well as the national-level population[69] and urbanization[70] projections of the SSPs as model input; and served as baseline "no SLR" projections (Supplementary Fig. 1). We must note that while a new version of GHS-POP became available in late 2022, we refrained from updating the results of this study to ensure consistency with the "no SLR" projections[51] used as a baseline.

### Modeling SLR-induced internal migration
To account for spatial migration patterns due to SLR, we produced spatial raster layers of submerged land per 10-year time step with the help of a bathtub approach, including all land with an elevation up to the amount of SLR in hydrological connection to the sea[129,130]. Following the methods described in ref. 131, we used the regionalized SLR projections of ref. 68 based on ref. 132, available at a spatial resolution of 2° by 2°. We adopted the median values (50th percentile) of RCP2.6, RCP4.5, and RCP8.5 from 2020 to 2100, with a mean SLR across the

Mediterranean at 0.31, 0.42, and 0.56 m by 2100 (relative to 1986–2005), respectively. Analyzing selected percentiles per SLR scenario[66,123,131,133–135] (and/or selected scenarios) is common practice in scenario-based modeling studies. The median SLR values used are in line with those reported in the IPCC's Special Report on the Ocean and Cryosphere in a Changing Climate (SROCC)[19]; however, compared to those reported in the IPCC's 6th Assessment Report (AR6), they are on the low end of the uncertainty range[54,136]. Therefore, we anticipate the number of internal migrants projected due to SLR to be rather conservative estimates than overestimations.

Next, we spatially attributed the SLR projections per RCP and decadal time step to the Mediterranean Coastal Database (MCD)[137], following the approach described in ref. 131. In the MCD, the Mediterranean coastline is split into roughly 12,000 coastal segments of varying length based on the physical and socioeconomic characteristics of the coast[138]. Accounting for the amount of SLR per coastal segment, we calculated submergence per 10-year time step with the help of the Multi-Error-Removed Improved-Terrain Digital Elevation Model (MERIT DEM), available at a horizontal resolution of 3 arc seconds (~90 m at the equator) and a vertical resolution of below 1 m (i.e., decimal values)[90]. We must note that we did not account for protection measures already in place due to a lack of consistent region-wide data[139]. We refrained from applying a region-wide or country-wide protection standard as it would have led to an overestimation of protection in those locations where no protection measures were present.

We included these submergence layers in CONCLUDE by classifying all submerged land as no longer available for human settlement for the respective scenario combination, thus ensuring that no population was allocated to these zones[15,16,37]. Further, we removed the population from the submerged raster cells based on the assumption that everyone living in these cells would migrate autonomously, and spatially distributed it to the remaining inhabitable cells according to the population potential. As these migration processes can take place over short distances, the same people may be forced to migrate multiple times in the course of the 21st century. The projections developed with this approach provided the 'no adaptation policies' reference scenarios, based on which the effects of each adaptation policy scenario on migrant numbers and spatial patterns of migration could be derived.

**Accounting for the effects of adaptation on internal migration**
**General approach.** We quantified the qualitative coastal adaptation SPAs (Fig. 1, Supplementary Text 1, and Supplementary Table 2) to model plausible locations where coastal adaptation measures would be implemented under the respective integrated scenario, accounting for hard protection, accommodation, and managed retreat strategies. Further, we developed distinct assumptions for the northern Mediterranean and the southern and eastern Mediterranean in order to reflect the differences in adaptive capacity[52,53]. These two regions broadly represented European Union (EU) and non-EU countries (Supplementary Table 1), characterized by the largest differences in socioeconomic development across the Mediterranean[51,140].

The raster-based adaptation layers produced in this step were harmonized to and processed at the spatial resolution of the MERIT DEM data (i.e., 3 arc seconds) before aggregating them to 30 arc seconds, the spatial resolution of CONCLUDE. In the aggregation process, we retained decimal values in hybrid raster cells, i.e., those cells partially located in the zone of the respective adaptation strategy. These adaptation layers (Supplementary Fig. 6) allowed us to run scenarios "with adaptation policies", from which we could then derive the effects of different adaptation policies on migrant numbers and spatial migration patterns by comparing the "with adaptation policies" results to the 'no adaptation policies' reference scenarios.

**Hard protection.** We assumed that hard protection measures would be implemented in each integrated scenario to some degree, the intensity of which was determined by the qualitative coastal adaptation SPAs (Supplementary Text 1 and Supplementary Table 2). We used the Dynamic Interactive Vulnerability Assessment (DIVA) modeling framework[62] to determine which coastal segments of the MCD[138] would be protected based on the socioeconomic characteristics in the MCD. To facilitate consistency with the input data of CONCLUDE, we extended the MCD with the population data of GHS-POP[107], calculating the number of people per coastal segment and elevation increment (based on MERIT DEM) with the help of zonal map statistics, based on the approach described in ref. 138. Protected segments in the MCD were established with a demand for safety function, assuming that protection levels are higher in the wealthier and more densely populated locations exposed to SLR-related coastal flooding (see ref. 62 for further detail). As modeling the spatial feedbacks between SLR, protection, and spatial population distributions, and feeding the updated population rasters into the MCD in each time step was beyond the scope of this study, we used the simplified assumption that protection measures would be built in 2020 and raised with SLR, with no additional protection measures built until 2100.

As DIVA was calibrated with data from the North Sea coast of the German province Schleswig-Holstein, where protection standards are high compared to most of the Mediterranean coast, the number and locations of protected segments produced by DIVA were implausibly high across scenarios. Consequently, we post-processed them (a) to reflect the protection standards of the Mediterranean context and (b) to ensure consistency with the coastal adaptation SPAs, thereby covering the uncertainty range related to the intensity of implemented protection measures per integrated scenario. Following the qualitative assumptions of the coastal SPA narratives (Supplementary Text 1 and Supplementary Table 2) and to ensure that only locations with high population densities would be protected under "Build with Nature" and "Save Yourself", we excluded all rural segments as well as those below a certain population density in the Low Elevation Coastal Zone (LECZ), which includes all land up to 10 m in elevation hydrologically connected to the sea[20]. In "Hold the Line", we applied the population density threshold only. Further, to reflect the lower adaptive capacity in the southern and eastern Mediterranean[52,53], we applied higher population density thresholds to these countries based on the assumption that hard protection measures are only initiated when more people are exposed to coastal flooding than in northern Mediterranean countries (see Supplementary Table 3 for the post-processing criteria).

The above procedure resulted in 2693 km of protected coastline in "Build with Nature", 1596 km in "Save Yourself", and 14,574 km in "Hold the Line", corresponding to 5, 3, and 27% of the total Mediterranean coastline, respectively (see Supplementary Fig. 6 for protection locations per integrated scenario). Although we did not account for the economic feasibility of protection measures, these numbers were roughly in line with previous work that used cost-benefit analysis to establish stretches of the coastline where protection would be economically robust: ref. 34 projected 3.4% and ref. 77 13.4% of the global coastline is worth protecting, with higher shares along the Mediterranean coastline[77]. Considering the concentration of population and assets in the immediate Mediterranean coastal zone[52], we deemed a share as high as 27% of the coastline to be protected under the "Hold the Line" scenario plausible. The coastal segments protected per integrated scenario are provided in Supplementary Data 2, which can be merged with the shapefiles of the MCD (available in ref. 137) to establish the location of hard protection measures per scenario (as shown in Supplementary Fig. 6). Based on the location of these protection measures, we were then able to mask out the land from the submergence layers that would not be submerged due to the presence

of hard protection (see Methods "Modeling SLR-induced internal migration").

**Accommodation.** We accounted for accommodation strategies by including setback zones (i.e., restriction of new development)[19] in coastal segments where no hard protection measures were implemented, following ref. 123. To define setback zones, we used a coastline buffer, following the Protocol on Integrated Coastal Zone Management (ICZM) in the Mediterranean[141] that entered into force in 2011, prescribing a distance of 100 m[118,121]. Due to the spatial resolution at which the data were processed (i.e., 3 arc seconds), we had to approximate the coastline buffer: we applied a coastline buffer of 6 arc seconds (~150 m in the Mediterranean) under "Build with Nature", based on the assumption that the ICZM Protocol would be followed and extended. In "Hold the line", we applied a buffer of 3 arc seconds, thereby roughly reflecting the requirements of the ICZM; we assumed that the ICZM Protocol would not be implemented in "Save Yourself" due to limited international cooperation and policy effectiveness.

In "Build with Nature", we assumed additional setback zones in areas at risk from regular coastal flooding (i.e., not protected by hard protection measures) to facilitate the restoration of coastal wetlands, thereby preserving their coastal protection function[19]. To account for regular flooding, we modeled the coastal floodplain from 2020 to 2100 using the bathtub approach, by adding the ESL height to the respective amount of SLR for each time step. Assuming that setbacks zones would be larger in the Mediterranean North compared to the South and East due to the higher adaptive capacity, we used the 50- and 25-year return periods to determine areas at risk from regular coastal flooding, respectively. Next, we combined the floodplain of each time step with coastal wetland data from the Global Lakes and Wetlands Database (GLWD) available at a horizontal resolution of 30 arc seconds[142]. We used the category 'Coastal Wetland (incl. Mangrove, Estuary, Delta, Lagoon)' only and selected those wetlands where the 2015 population density was lower than 300 people km$^{-2}$, following ref. 133. At higher population densities, we assumed that wetlands would not have sufficient accommodation space to migrate inland with rising sea levels[133,143]. Further, we complemented the GLWD data with high-resolution (3 arc seconds) spatial data of salt marshes, available for countries in the northern and eastern Mediterranean[144]. We added a buffer of 6 arc seconds to each coastal salt marsh to account for the accommodation space needed, following the requirements of the ICZM protocol[120,141]. Last, we combined the coastal floodplains and wetlands data with each other to remove those wetlands not located in the coastal floodplain, and combined them with the coastline buffer to produce the final setback zones per time step. We then integrated the setback zones in CONCLUDE by classifying all land that fell into these zones per time step as no longer available for human settlement, thereby inhibiting migration towards these zones.

**Managed retreat.** In "Build with Nature", we additionally accounted for proactive managed retreat in unprotected and frequently flooded locations, defined as the 2-year return period in the southern and eastern Mediterranean and the 5-year return period in the northern Mediterranean, reflecting the differences in adaptive capacity across the two regions. Previous work used the 10-year return period for establishing managed retreat zones[34], which we found implausibly large for this study based on visual inspection. Using the bathtub approach, we modeled these retreat layers per time step and included them in CONCLUDE, moving the population from the retreat zones to the remaining inhabitable locations according to the population potential.

**Calculation of migration hotspots**
To estimate the robustness of our results in a spatially explicit manner, we calculated cell-based migration hotspots for each scenario based on the upper/lower 10% of all raster cells that experienced in- and out-

migration by 2100, respectively (compared to the "no SLR" baseline projections). Following the approach of refs. 15,16, we combined hotspots across the three SSP-RCP combinations, both without adaptation policies and with the implementation of adaptation policies (i.e., six in total), thereby establishing raster cells where at least five scenarios projected high levels of in- or out-migration ("high confidence"), where three to four scenarios agreed ("medium confidence"), and where two or fewer scenarios agreed ("low confidence").

**Reporting summary**
Further information on research design is available in the Nature Portfolio Reporting Summary linked to this article.

## Data availability
All input data to CONCLUDE produced in this study are based on publicly available data sources as described in the Methods and are available from the corresponding references. The stretches of coastline protected with hard protection measures per integrated scenario, generated with the DIVA modeling framework, are provided in Supplementary Data 2.

## Code availability
The model code for producing the spatial migration projections is available from ref. 145 and https://github.com/lena-reimann/CONCLUDE.git.

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

## Acknowledgements

This work was initiated as part of a Fulbright doctoral scholarship that allowed L.R. to work at the CUNY Institute for Demographic Research (CIDR) during a 6-month research visit. Further, L.R. and J.C.J.H.A. were

supported by the ERC-funded project COASTMOVE (grant 884442, www.coastmove.org). A.T.V. was supported by the CoCliCo project, funded by the European Union's Horizon 2020 research and innovation program (grant 101003598).

## Author contributions

L.R., B.J., and A.T.V. designed the research. L.R. developed the scenario assumptions with support from B.J. and produced the modeling results. C.W. provided the data on hard protection measures. L.R. and N.B. analyzed the results. J.C.J.H.A. contributed to the contextualization of the results. L.R. drafted the manuscript. All authors reviewed and edited the manuscript.

## Funding

## Competing interests

The authors declare no competing interests.
