## [Peer Review File · Nature Communications]

Exploring spatial feedbacks between adaptation policies and internal migration patterns due to sea-level riseREVIEWER COMMENTS

Reviewer #1 (Remarks to the Author):

This paper has a really good idea. I have several questions and concerns, however, that preclude immediate publication.

These are not in the order they appear in the manuscript:

- On line 38 the authors use the word "consolidate" (These spatial patterns are expected to consolidate in the future...). I'm not sure exactly what the authors mean.
- On line 40 the authors write "Therefore, SLR will gradually result.." this is perhaps too strong of a word. Of the climate impacts with the greatest potential to spur human migration, I think SLR is probably the greatest, but its definitely not "will result" in migration.
- On line 43, the authors use "Lincke and Hinkel" to set up their argument. But why highlight this one estimate? There are more than a dozen global estimates regarding submergence of populations due to SLR. Why choose this one? Furthermore, how do the authors rectify this estimate -- of 40 million by 2100 with 1.1M of SLR globally -- with your result of 20 million in just the Mediterranean under, what, 0.6m at the most? I think there's also something lost in translation somewhere as line 46 talks about 45 million migrants by 2100 with the 1-in-10 year floodplain. Just 5 million more than the 40 million? I suggest the authors (1) double check these numbers and (2) at least include a broader set of estimates besides just Lincke and Hinkel. Or at least justify the focus of this one article.
- On line 64, the authors discuss the levee effect and write that it will result "in an increase in exposure." How can the levee effect increase exposure? Or is this just a general increase in potential vulnerability? How can those behind levees experience an increase in exposure? Maybe they might have an increase in exposure if a levee fails? Based on my readings of the vulnerability literature, I'm not sure I would conclude this.
- Regarding Figure 1. This figure and the results are a big problem right now. The authors are grouping together adaptation strategies that operate in opposing directions, which muddles their results. protection, buffers, and coastal restoration are designed to hinder migration, to keep people in place, to prevent migration. But managed retreat is the exact opposite, its designed to enhance migration, to move people, to facilitate migration.

This is especially difficult to interpret their results with these categories. Including all three types together, we end up with apples-to-oranges comparisons.

I'd suggest the authors *strongly* consider breaking out adaptation policies from the SSPs to evaluate the impact each adaptation type has on migration. Otherwise, it is incredibly difficult to interpret these results.

Furthermore, these labels for the SPAs are useful and reflect the SSPs but don't line up with the adaptation policies themselves. Nor am I seeing how "autonomous" migration is comparable to the four main adaptation polices included.

I'm also not entirely sure I would consider a coastline buffer or restoration to be 'accommodation' strategies. Those are more akin to soft armoring. Accommodation is 'living with higher water levels', things like elevating roads, switching to saline-tolerant crops, etc. These are not really accommodation strategies.

I also don't see how 'build with nature' includes managed retreat. Why can't we have managed retreat under SSP5, where adaptation costs are low?

In summary on Figure 1, I suggest the authors (1) break out the adaptation policies to evaluate the

impact each adaptation type has on migration. (2) Pick other policies that are more comparable to the four main adaptation policies included. "autonomous" migration isn't really consistent with any of them. (3) Think of modelling other accommodation strategies. and (4) Include managed retreat under SSP5.

-On line 92 the authors write that SLR "may result in over 20 million internal migrants by 2100." Honestly, I had a hard time connecting Figure 1 to this figure. A table would be extremely useful here. I'm having a hard time understanding the 20 million migrants by 2100 and the comparisons using Figure 2.

- Regarding Figure 2. I'm having a hard time properly interpreting these results. Do they suggest that nearly 4 million people will migrate in the Mediterranean region in the 2020s? Holy cow that's alot!

I also don't understand how SSP1 has the highest migrants under all scenarios, yet RCP2.6 has the lowest amount of SLR?

I also don't see how the curves in panels B are possible. How can there be a cascade effect in 2070 or 2090? Nothing in the model suggests a cascade effect can occur since the migrant populations are simply redistributed throughout the region. Is the gravity modeling impacting migrants too? The methods section doesn't specify this. Can the authors comment on how such a cascade is possible in their model? Otherwise, I find these results implausible.

- On line 141 the authors write "most migration takes place out of a coastal strip submerged by SLR into inland locations." Is this because of the modelling choice in splitting out the urban and rural areas? I'm trying to get a sense of how much of this might be due to results that reflect the model assumptions and results that might not deterministically arise from the assumptions. I can't tell this from the methods section as its currently written. I ask because coastal populations continue to swell and some models suggest coastal areas will continue to be destinations for migrants far into the future. I'm trying to rectify this finding with other findings in the literature.

- Regarding Figure 4. This figure is really difficult to interpret. I see almost no green in the figure. The little green there is seems located in coastal areas, rather than inland areas. Which is the opposite of the conclusion on the previous page.

Since the gray color includes both positive and negative values, its hard to tell where there is out-migration and where there is in-migration. Taking a charitable view of the results, it would appear that most population redistribution occurs in exceptionally rural areas? How can that possibly be when almost all migration over the past century (and expected this century) is to be rural-to-urban or urban-to-urban. Urban-to-rural and rural-to-rural migration (globally) comprise fractions of migration streams but these results suggest tremendous migration into rural areas? How do the authors rectify this finding?

Also, am I correct in reading that Paris and Madrid -- completely landlocked cities -- LOSE population because of SLR? Otherwise why are they pink ("Pink colors show out-migration, green colors show in-migration"). Perhaps the authors could clarify how a sea-level rise migration model can spur migration out of these areas that are unaffected by sea-level rise.

Also, why is the scale labelled "uninhabited"?

- On line 188, the authors write about their robustness checks. I don't find this "robustness check" particularly compelling. All of these modelled scenarios have some form of uncertainty inherent to them but it looks like the only modeling done is the 50th percentile. So IF SLR is exactly the 50th percentile value and IF the SSP follows exactly the 50th value, then the results suggest "X". But this

approach doesn't tell us the true amount of uncertainty inherent to the model here. Simply counting how many scenarios are within specified upper/lower bounds isn't enough, in my opinion. I'd suggest the authors try and better incorporate uncertainty due to (1) the sea-level rise scenarios and (2) INCLUDE.

- Regarding Figure 5. This figure is super confusing compared to Figure 4. The Nile delta is very red, suggesting high confidence in in-migration, but in Figure 4, most of the delta is pink (suggesting OUT migration). Perhaps the authors could clarify these figures? Its just very hard to interpret what is occurring. How can the authors have high confidence for in-migration in many areas but show out-migration in figure 4?

- On line 208 the authors write "the implementation of adaptation policies is largely driven by national to local governments." What level of government is excluded from this sentence? I suggest the authors edit this.

- On line 263 the authors write "we covered the uncertainty range of future SLR." But not the uncertainty of the ranges of future SLR. Which is important because both the population scenarios in the SSPs, the SLR scenarios in the RCPs, and the downscaling from INCLUDE have TONS of uncertainty. The authors have not modelled those levels of uncertainty.

- On line 283, the authors write a long sentence describing all the different ways CONCLUDE has been used. This entire sentence is a logical fallacy (appeal to popularity). Simply because it has been used previously doesn't mean its any good for this study. I don't care whos used it in the past (and maybe I disagree with their uses, maybe I don't), I just care about its fitness for use in THIS study. And this sentence does not convey that. I suggest the authors delete this sentence.

- One line 291, the authors write about "attractiveness" But they don't use economic attractiveness. This seems like a major exclusion. Can the authors justify this exclusion?

- On line 296, the authors write about A_i . What goes into A_i is buried in the Supplementary material. I suggest the authors include it here.

- On line 309, the authors write about the 2 degree by 2 degree grid from Kopp et al. 2 degrees is 7200 arc seconds, right? This, what, a 2000 meter resolution? But the MERIT DEM is 3 arc seconds and 90 m resolution. The ICZM is applied at 3, 4, and 6 arc seconds or 150 meters. The GLWD is 30 arc seconds or 900 meters. I don't think the GHS-POP resolution was mentioned in the paper. Because the spatial resolutions of all the various input data is so vastly different, I'm not sure exactly what to conclude. I'd like to hear a justification from the authors about why this isn't an ecological fallacy.

- On line 310, the authors write the SLR amounts. These are only the 50th percentile values. What about the 80th percentile values? I went ahead and looked at the citations for these and the uncertainty intervals are provided in both citations (Table 1 in both). Why isn't the uncertainty in SLR propagated through your estimates? Furthermore, how in the world does SSP1 produce the most migrants when it has the least SLR?

- On line 345, the authors write that they "extended with the population data of GHS-POP." I'm unfamiliar with these datasets so I'm unsure exactly how the authors "extended the population data of the GHS-POP". Perhaps the authors could explain how they did this.

- The sentence on line 347 (starting with "Protected segments were established..."), This whole section needs to be spelled out. How did you establish a safety function? How did wealth increase in these areas? Did you model wealth? (I get how pop density might since you model population). This is entirely too ambiguous.

- The sentence on line 350 starting with "As the number and locations of protected segments based on the DIVA results were similar across scenarios...", This is vaguely written and I'm unsure what's happening. DIVA basically said the scenarios don't matter so you post-processed them? How did you post-process them? This is entirely too ambiguous.

- On line 355, the authors write that they "excluded all rural segments..." Why? Justify this assumption.

- On line 361, the authors write about the km of protected coastline. Are there cost estimates for armoring that are per km of coastline to get a sense of how reasonable these assumptions are? I have no idea if 14,000 km of protected coastline is even feasible. What if it costs like \$1 trillion?

3-27% is seems unreasonably wide, but, again, I have no real way of determining if this is appropriate based on the text as its currently written.

- Line 367, The "Accommodation" section. I'm not sure I would call any of this accommodation. Setback zones and coastal restoration is usually referred to as a "soft armoring". Accommodation might be deploying pumps, or changing to more saline tolerant crops. Those accommodation strategies are absent in this analysis. I'd suggest either dropping this part and rerunning your analysis or actually including some real accommodation strategies.

- On line 372, the authors write that the ICZM has a distance of 100m but then the authors use a buffer of 6 arc seconds or 150 m. 150m >100m. Why choose 6 arc seconds? Why choose a buffer that's 50% larger than the statutory buffer?

-- On line 378, the authors write about the "we assumed additional setback zones in areas at risk from regular coastal flooding to facilitate the restoration of coastal wetlands." Is the entire Mediterranean coastal wetlands? I'm unfamiliar with the local geography of the Med region. I don't know if this is a good assumption. And the authors aren't helping me along. Presumably, the entire coastline isn't coastal wetlands but will some areas that _aren't_ wetlands be included in this buffer?

- On line 380, the authors write about adding "the ESL height to the respective amount of SLR for each time step." Why is this mentioned here but not in the modelling SLR section? Where does this come from? The Kopp citation?

- On line 383, the authors describe the 25- and 50-year Return Periods as "reflecting the differences in adaptive capacity between these regions." But I fail to see how how extreme sea levels or even return levels associated with flooding are related to adaptive capacity? Perhaps the authors can explain this?

- On line 394, the authors write that they make some areas "no longer available for human settlement, without removing population from these cells." Why? I think this is the only assumption that purposefully keeps people in place? What's the justification?

- On line 398, the authors introduce the use of a 2-year return period in the southern and eastern Mediterranean and the 5-year return period in the northern Mediterranean. But why? There is (1) no justification provided for why they chose these specific return periods and (2) why they are different for each region. Reading between the lines with charity, its possible the authors are alluding to the "differences in adaptive capacity between these regions" on line 383. But again, I can't see how return periods are related to adaptive capacity.

- Regarding the code availability statement. This is completely unacceptable in 2022. Please provide the code to understand what you are doing. Especially since the assumptions are poorly laid out. "available upon request" basically signifies the code is not available in practice.

Reviewer #2 (Remarks to the Author):

An interesting paper adding to the canon of work on possible migration futures occurring in response to sea level rise. I found the methodology robust within the demography paradigm by using a spatially-explicit gravity-based model. If I had a criticism of this paper it would be around whether the assumptions of gravity-based models will hold into the future, for example, whether 'city islands' ringed by sea defences will continue to act as attractors and whether the bigger motivation for migration is more likely to be repeated storm surge events and decreasing returning displaced populations (e.g. see Fussell, E., Sastry, N. and Van Landingham, M., 2010. Race, socioeconomic status, and return migration to New Orleans after Hurricane Katrina. *Population and environment*, 31(1), pp.20-42). Thus, to my reading the paper neglects to question assumptions around the way in which people are envisaged to respond to climate stresses. This type of model is useful for modelling slow demographic change but has some important limitations in modelling migration flows and the multiple and novel ways climate change might affect migration. So in summary a solid piece of work that maybe could be more reflexive over the assumptions of the model.

Reviewer #3 (Remarks to the Author):

This paper presents an interesting set of findings examining a calibration of a gravity model called CONCLUDE to the Mediterranean region, coupled with a model of SLR out to 2100 and a set of adaptation scenarios, to examine the effects of hard adaptation measures on coastal outmigration, differentiated between the comparatively rich North Mediterranean and the comparatively poor South-East Mediterranean.

The paper is clear and well structured. I will not provide a list of specific criticisms. I outline only two potential flaws that might stand in the way of it being a valid and important contribution. Both issues relate to the reliance on scenarios as the key experiment/shock.

First, migration is already implicit within the SSPs. Your methods don't discuss this, and leave out the migration element when describing the SSPs applied to your model. To be valid, you need to demonstrate how allowing migration to be endogenous to your model – and possibly heterogeneous across place – is not inconsistent with the SSPs. In particular, you should demonstrate that the migration flows that emerge in your simulation do not violate their assumed states in the SSPs (or importantly, violate them differently across your rich and poor regions). (Doing this well, in my mind, makes it an even more important contribution, as a roadmap for future researchers to see better how to do this – draw on the SSPs as backbones for modeled processes that are at least partially endogenous to them).

Second, it is a mis-statement to say that your contribution is the first to capture feedbacks between among population, adaptation policy and SLR, because only population is endogenous to your model. SLR and adaptation policy are fixed by your scenarios, and are thus providing bounding conditions for population to evolve, nothing more. The only mechanism through which feedbacks can happen in your model are the basic premises of the gravity model (bigger cities pull more) which are more than a century old and are not your contribution. If adaptations were an endogenous response to population change (as you might get from an ABM), this could possibly be part of a feedback, but to my understanding, this is not what you have modeled. Thus, the feedback language needs to be removed and the contribution clarified, or a clear demonstration of the feedback (i.e., showing in a causal loop diagram a set of endogenous processes in your model that complete a feedback loop and include the novel adaptation and SLR features of your model) needs to be provided.

I look forward to a version of this manuscript that addresses the above issues, as I believe it will be quite a well-read contribution to the literature.

REVIEWER COMMENTS

We would like to thank the three reviewers for their thoughtful and constructive comments. To address their comments, we have substantially revised the manuscript along four main aspects, 1) we have revised the introduction section to clearly point out the aims of the study; 2) we have revised the results section, particularly the figures, to improve readability; 3) we now discuss the model limitations in more detail; and 4) we provide additional detail in the methods section to transparently describe the steps taken for modeling adaptation policies and migration. All changes made in the manuscript are shown as track changes. We believe that we have addressed the reviewers' concerns and hope that the manuscript is now acceptable for publication.

Reviewer #1 (Remarks to the Author):

This paper has a really good idea. I have several questions and concerns, however, that preclude immediate publication.

We would like to thank Reviewer #1 for the constructive comments. We have substantially revised the manuscript to address the points raised. Based on the reviewer's comments, we realized that we have not stated the main aim of the study clearly enough, which is the exploration of different adaptation policies developed on the basis of the Shared Policy Assumptions (SPAs) in the context of the SSP-RCP-SPA scenario framework. We have reworked the introduction section to clarify this aspect; we have revised the results section for readability; we now reflect on the model limitations in more detail; and we provide additional detail on scenario assumptions and corresponding data processing in the methods section. Furthermore, we have conducted additional analysis based on new model runs to support the reasoning behind using integrated adaptation policy scenarios. We believe that we provide sufficient detail now to convey the study aim, key messages, and limitations clearly. Please see our point by point responses to each comment below.

These are not in the order they appear in the manuscript:

1. On line 38 the authors use the word "consolidate" (These spatial patterns are expected to consolidate in the future...). I'm not sure exactly what the authors mean.

We have rewritten the sentence to make it clearer. It now reads "These spatial patterns are expected to continue in the future due to progressing urbanization as well as continued high attractiveness of coastal areas for human settlement" (lines 40-42).

2. On like 40 the authors write "Therefore, SLR will gradually result.." this is perhaps too strong of a word. Of the climate impacts with the greatest potential to spur human migration, I think SLR is probably the greatest, but its definitely not "will result" in migration.

We have rephrased the sentence: "Therefore, SLR may result in permanent migration due to slow-onset impacts such as submergence of land or coastal erosion as well as in temporary displacements during coastal flooding due to ESL" (lines 42-44).

3. On line 43, the authors use "Lincke and Hinkel" to set up their argument. But why highlight this one estimate? There are more than a dozen global estimates regarding submergence of populations

due to SLR. Why choose this one? Furthermore, how do the authors rectify this estimate -- of 40 million by 2100 with 1.1M of SLR globally -- with your result of 20 million in just the Mediterranean under, what, 0.6m at the most? I think there's also something lost in translation somewhere as line 46 talks about 45 million migrants by 2100 with the 1-in-10 year floodplain. Just 5 million more than the 40 million? I suggest the authors (1) double check these numbers and (2) at least include a broader set of estimates besides just Lincke and Hinkel. Or at least justify the focus of this one article.

We would like to thank the reviewer for pointing out these inconsistencies. We have now revised the paragraph and have double-checked and corrected the numbers. We have also expanded the paragraph to broaden the set of studies (lines 45-62).

4. On line 64, the authors discuss the levee effect and write that it will result "in an increase in exposure." How can the levee effect increase exposure? Or is this just a general increase in potential vulnerability? How can those behind levees experience an increase in _exposure_? Maybe they might have an increase in exposure if a levee fails? Based on my readings of the vulnerability literature, I'm not sure I would conclude this.

Thank you for this comment. We are aware that different definitions and uses of the terms 'exposure' and 'vulnerability' can be found in the literature, especially across disciplines. Here we use the IPCC's AR6 definition of exposure, which is "[t]he presence of people; livelihoods; species or ecosystems; environmental functions, services, and resources; infrastructure; or economic, social, or cultural assets in places and settings that could be adversely affected". Therefore, exposure refers to the presence of people etc. in potentially flooded locations independent from the presence of adaptation measures; these are encapsulated in the concept of vulnerability, defined as "The propensity or predisposition to be adversely affected. Vulnerability encompasses a variety of concepts and elements including sensitivity or susceptibility to harm and lack of capacity to cope and adapt." (IPCC 2022 Annex II).

In this context, the levee effect may result in increased exposure as protected areas become more attractive for human settlement (lines 70-74). The levee effect, and the increase in urban areas behind levees, has been discussed in literature (e.g. Di Baldassarre et al 2018) and several cases studies show the empirical evidence of increasing exposure and higher protection standards (de Moel et al 2011).

5. Regarding Figure 1. This figure and the results are a big problem right now. The authors are grouping together adaptation strategies that operate in opposing directions, which muddles their results. protection, buffers, and coastal restoration are designed to hinder migration, to keep people in place, to prevent migration. But managed retreat is the exact opposite, its designed to enhance migration, to move people, to facilitate migration.

This is especially difficult to interpret their results with these categories. Including all three types together, we end up with apples-to-oranges comparisons.

I'd suggest the authors *strongly* consider breaking out adaptation policies from the SSPs to evaluate the impact each adaptation type has on migration. Otherwise, it is incredibly difficult to interpret these results.

Furthermore, these labels for the SPAs are useful and reflect the SSPs but don't line up with the adaptation policies themselves. Nor am I seeing how "autonomous" migration is comparable to the

four main adaptation policies included.

I'm also not entirely sure I would consider a coastline buffer or restoration to be 'accommodation' strategies. Those are more akin to soft armoring. Accommodation is 'living with higher water levels', things like elevating roads, switching to saline-tolerant crops, etc. These are not really accommodation strategies.

I also don't see how 'build with nature' includes managed retreat. Why can't we have managed retreat under SSP5, where adaptation costs are low?

In summary on Figure 1, I suggest the authors (1) break out the adaptation policies to evaluate the impact each adaptation type has on migration. (2) Pick other policies that are more comparable to the four main adaptation policies included. "autonomous" migration isn't really consistent with any of them. (3) Think of modelling other accommodation strategies. and (4) Include managed retreat under SSP5.

Thank you for raising these important issues, which we respond to along the four main points above.

(1) We decided not to focus on analyzing adaptation measures separately, but we have taken the reviewer's comment very seriously and have conducted additional analysis to underpin our response. We also revised the manuscript as the reviewer rightfully addressed some unclear issues. Our line of arguments is as follows:

As the main aim of this study is to explore the effects of different adaptation policies on migration due to sea-level rise (SLR), we followed the integrated scenario approach using the RCP-SSP-SPA scenario framework by van Vuuren et al (2014). To do so, we have developed Shared Policy Assumptions (SPAs) for coastal adaptation that are plausible under the socioeconomic developments described in the respective Shared Socioeconomic Pathway (SSP). Thus, we aim at addressing adaptation policies that encompass a range (or combination) of measures that fit the SSPs, and therefore a more realistic view of how adaptation will take place. In addition, these integrated scenarios cover the uncertainty range of different integrated adaptation policies and their effects on SLR-related migration. For instance, we do not consider an SPA that solely relies on managed retreat to be plausible under any SSP as large shares of the coastal population would have to migrate, in particular out of urban locations that will likely receive some degree of hard protection (Lincke and Hinkel 2018, 2021); the same holds true for the implementation of setback zones (Lincke et al 2020).

However, following the reviewer's comment, we conducted additional analysis to further illustrate this point. Hereby, we provide an example where we have conducted additional model runs and analysis, exploring the effect of each individual adaptation measure on migration numbers and spatial patterns until 2100, with a focus on Greece. Figure 1 shows that implementing setback zones would lead to over 1.2 million migrants in Greece alone, resulting from the number of autonomous migrants (i.e. 884,861) plus roughly 317,000 people who are restricted from migrating into these zones. Implementing managed retreat measures along the entire coastline would lead to over one million migrants, while hard protection measures would result in about 657,000 migrants until 2100. The combination of the three adaptation measures in 'Build with Nature' results in a plausible intermediate number of migrants (about 713,000) that is lower than implementing managed retreat and setbacks alone or not implementing any adaptation measures at all ('autonomous migration'), due to the protection of densely populated locations. This example supports our reasoning for using an integrated 'Build with Nature' SPA as it results in plausible migration numbers (as opposed to those resulting from individual

measures), while being consistent with the socioeconomic developments described under the corresponding SSP (i.e. SSP1).

Figure 1 Number of migrants per adaptation strategy under SSP1-RCP2.6 in Greece until 2100

Furthermore, we would like to stress that the migration outcomes of each integrated scenario (such as 'Build with Nature') are meant to be compared to the respective outcomes without any adaptation policies ('no adaptation policies', e.g. SSP1 – RCP2.6) rather than directly comparing the three integrated scenarios with each other. We have noticed, however, that this aspect of our study has not been stated clearly in the manuscript. Therefore, we added text to the introduction (lines 87-101) as well as the methods (lines 338-342) sections for clarification of these points.

- (2) We are not entirely sure whether we understand this suggestion correctly. To our understanding, the assumptions regarding the adaptation measures pursued per SPA align well with the developed SPA narratives (Supplementary Text 1) and the basic SSP assumptions (Supplementary Table 1). In scenario analysis, there is always a baseline scenario (in this case 'no adaptation policies') to which the scenarios with adaptation policies are compared. Therefore, the term 'autonomous migration' does not refer to a specific adaptation policy, but rather to a baseline scenario without adaptation policies; thus, we assume that the population in locations submerged due to SLR will migrate (i.e., we do not consider trapped populations). However, it might be the case we were not clear in our definitions, and we have further clarified our approach in lines 45-47, 216-220, and 392-393.
- (3) Indeed, there are different definitions of accommodation. We followed Oppenheimer et al. (2022) where setback zones are one type of accommodation measures; we have added the corresponding reference in the text (line 466). We refrained from accounting for additional accommodation strategies such as raising of roads or buildings or early warning systems as these have a limited effect on migration (as rightly pointed out by the reviewer), which would be very localized, and cannot be captured in this continental-scale modeling study. Further, we would like to rectify that the setback zones (i.e., the coastline buffer combined with coastal wetland restoration) as implemented in the model do not hold people in place, but restrict new development, i.e., people cannot migrate towards these zones, while they can move away from these locations (see lines 499-500). Therefore, setback zones increase the number of migrants in locations where we would otherwise see high in-migration into coastal locations.
- (4) We agree with the reviewer that managed retreat may be a plausible adaptation strategy under SSP5 due to high economic growth, which leads to low adaptation challenges. However, managed

retreat is not a plausible adaptation strategy under the 'Hold the Line' SPA that assumes highly engineered solutions to be pursued (Supplementary Text 1). Since we combine SSP5 with the 'Hold the Line' SPA, we do not include managed retreat in this integrated scenario.

6. On line 92 the authors write that SLR "may result in over 20 million internal migrants by 2100." Honestly, I had a hard time connecting Figure 1 to this figure. A table would be extremely useful here. I'm having a hard time understanding the 20 million migrants by 2100 and the comparisons using Figure 2.

We would like to thank the reviewer for raising this issue. We have reworked the text to help the reader navigate between the text and Figure 2 (lines 104-135).

7. Regarding Figure 2. I'm having a hard time properly interpreting these results. Do they suggest that nearly 4 million people will migrate in the Mediterranean region in the 2020s? Holy cow that's alot!

I also don't understand how SSP1 has the highest migrants under all scenarios, yet RCP2.6 has the lowest amount of SLR?

I also don't see how the curves in panels B are possible. How can there be a cascade effect in 2070 or 2090? Nothing in the model suggests a cascade effect can occur since the migrant populations are simply redistributed throughout the region. Is the gravity modeling impacting migrants too? The methods section doesn't specify this. Can the authors comment on how such a cascade is possible in their model? Otherwise, I find these results implausible.

Assuming proactive adaptation that involves managed retreat and setback zones being implemented in 2020, the 'Build with Nature' scenario would indeed lead to 3.1 million migrants in 2020 (lines 113-114). However, we would like to emphasize that we do not aim to predict the number of migrants in the course of the 21st century, but that we rather explore the plausible range of migrant numbers if such adaptation policies were pursued (see also our response to comment 5 (1)).

The high number of migrants under SSP1 are indeed the result of the relatively low amounts of SLR, but in combination with the underlying SSP assumption that urban areas in the coastal zone are very attractive and exert a pull on migrants. As urban settlements are concentrated along the Mediterranean coastline, a large share of migrants is distributed in the immediate coastal zone, which is affected by submergence due to SLR in the next time step(s), thus leading to high migrant numbers until 2100. This effect is less pronounced under SSP5 where urban sprawl is high and under SSP3 where urbanization rates are low. The similar total number of migrants across SSP-RCP combinations (no adaptation) further shows that socioeconomic developments as described in the SSPs have a larger influence on the number of migrants compared to the amount of SLR. We have added text (lines 185-196) to reflect upon this effect.

The cascade effect seen in Figure 2b ("tipping point" in the manuscript) results from the acceleration of SLR in the second half of the 21st century, which occurs earlier under RCP4.5 and RCP8.5 compared to RCP2.6 (lines 122-125). As we assume the population residing in a submerged raster cell to migrate, this effect is driven by the timing when densely populated raster cells are submerged due to SLR. We are now reflecting upon the caveats of the modeling approach in more detail in lines 235-251.

8. On line 141 the authors write "most migration takes place out of a coastal strip submerged by SLR into inland locations." Is this because of the modelling choice in splitting out the urban and rural areas? I'm trying to get a sense of how much of this might be due to results that reflect the model assumptions and results that might not deterministically arise from the assumptions. I can't tell this from the methods section as its currently written. I ask because coastal populations continue to swell and some models suggest coastal areas will continue to be destinations for migrants far into the future. I'm trying to rectify this finding with other findings in the literature.

Thank you for raising this point. Please see lines 389-394 for the method used for modeling migration due to submergence by 'picking up' the submerged population and spatially distributing it based on the population potential of the remaining inhabitable cell. This is done separately for urban versus rural areas (lines 365-367). Indeed, the observed spatial patterns reflect these model assumptions.

Regarding coastal population growth - to our knowledge - no continental- to global-scale studies that account for the impacts of SLR on spatial population distributions (hence migration) have been published thus far. As far as we are aware, studies projecting coastal population growth (e.g. Nicholls 2004, Neumann et al 2015, Merkens et al 2016, Reimann et al 2018a, 2021) do not account for these spatial feedbacks. One national-scale example that projects coastal population growth despite progressing SLR in Bangladesh (Bell et al 2021) differs from our study in two major aspects, which inhibits direct comparisons of results: 1) it accounts for coastal flooding due to extreme sea levels (ESL) rather than submergence due to SLR; 2) it uses an agent-based approach, modeling migration between districts rather than on the raster cell level. At the coastal district level (or another coastal definition such as the Low Elevation Coastal Zone, LECZ), we may also project population growth until 2100; however, this analysis was beyond the scope of this study. We have added text (lines 235-251) to reflect upon the modeling assumptions used in this study.

9. Regarding Figure 4. This figure is really difficult to interpret. I see almost no green in the figure. The little green there is seems located in coastal areas, rather than inland areas. Which is the opposite of the conclusion on the previous page.

Since the gray color includes both positive and negative values, its hard to tell where there is out-migration and where there is in-migration. Taking a charitable view of the results, it would appear that most population redistribution occurs in exceptionally rural areas? How can that possibly be when almost all migration over the past century (and expected this century) is to be rural-to-urban or urban-to-urban. Urban-to-rural and rural-to-rural migration (globally) comprise fractions of migration streams but these results suggest tremendous migration into rural areas? How do the authors rectify this finding?

Also, am I correct in reading that Paris and Madrid -- completely landlocked cities -- LOSE population because of SLR? Otherwise why are they pink ("Pink colors show out-migration, green colors show in-migration"). Perhaps the authors could clarify how a sea-level rise migration model can spur migration out of these areas that are unaffected by sea-level rise.

Also, why is the scale labelled "uninhabited"?

We agree that the maps shown in Figure 4 in the previous version of the manuscript were difficult to read. We have now reworked the figure, zooming into one region per scenario (with the full maps in the Supplementary Information) and refining the color scale to clearly differentiate out-migration

from in-migration. To avoid confusion regarding the migration patterns shown in the figure, we have added two rows that present migration patterns for the 'no adaptation policies' (upper row) and 'with adaptation policies' (middle row) scenarios to complement the migration patterns that were previously presented based on a comparison of the 'no adaptation policies' and 'with adaptation policies' scenarios (lower row). We have also reworked the text describing Figure 4 (lines 152-182). Furthermore, we would like to point out that the light grey areas in the map represent uninhabited land as classified in the figure legend.

10. On line 188, the authors write about their robustness checks. I don't find this "robustness check" particularly compelling. All of these modelled scenarios have some form of uncertainty inherent to them but it looks like the only modeling done is the 50th percentile. So IF SLR is exactly the 50th percentile value and IF the SSP follows exactly the 50th value, then the results suggest "X". But this approach doesn't tell us the true amount of uncertainty inherent to the model here. Simply counting how many scenarios are within specified upper/lower bounds isn't enough, in my opinion. I'd suggest the authors try and better incorporate uncertainty due to (1) the sea-level rise scenarios and (2) INCLUDE.

In line with the overall aim of this study, the aim of the robustness check is to account for the uncertainty related to the results produced across all adaptation policy scenarios (including the lack of adaptation policies) and to establish those locations where all scenarios project a high share of in- or out-migration, referred to as 'migration hotspots' (lines 225-232). As the amount of SLR does not affect future migration substantially (see also response to comment 7), we are convinced that the robustness checks would not yield different results if a wider range of SLR projections was accounted for. Please see lines 235-251 for reflections on the uncertainties related to the modeling approach used.

11. Regarding Figure 5. This figure is super confusing compared to Figure 4. The Nile delta is very red, suggesting high confidence in in-migration, but in Figure 4, most of the delta is pink (suggesting OUT migration). Perhaps the authors could clarify these figures? Its just very hard to interpret what is occurring. How can the authors have high confidence for in-migration in many areas but show out-migration in figure 4?

We have reworked Figure 4 to clarify the points raised in comment 9 and are positive that we could resolve the confusion about the spatial patterns that were shown in the previous version of the figure. For Figure 5, we decided to zoom into two regions, the western and eastern Mediterranean (similar to the previous version of Figure 4), to increase readability of the maps. We have moved the full map as an additional panel to Supplementary Figure 5.

12. On line 208 the authors write "the implementation of adaptation policies is largely driven by national to local governments." What level of government is excluded from this sentence? I suggest the authors edit this.

We rephrased the sentence. It now reads "However, the implementation of adaptation policies is largely realized at national and subnational level and is driven by the overall adaptive capacity per country, which varies considerably across the Mediterranean" (lines 260-262).

13. On line 263 the authors write "we covered the uncertainty range of future SLR." But not the uncertainty of the ranges of future SLR. Which is important because both the population scenarios in

the SSPs, the SLR scenarios in the RCPs, and the downscaling from INCLUDE have TONS of uncertainty. The authors have not modelled those levels of uncertainty.

As alluded to in our response to comment 10, our aim was not to account for the uncertainty range within SLR projections (i.e. using different percentiles) due to the limited effect on migration patterns. We, however, cover the uncertainty range related to future SLR, socioeconomic development, and adaptation policy scenarios in our scenario selection, analyzing three integrated scenarios along with their respective 'no adaptation policies' counterparts (lines 313-317; 327).

14. On line 283, the authors write a long sentence describing all the different ways CONCLUDE has been used. This entire sentence is a logical fallacy (appeal to popularity). Simply because it has been used previously doesn't mean its any good for this study. I don't care whos used it in the past (and maybe I disagree with their uses, maybe I don't), I just care about its fitness for use in THIS study. And this sentence does not convey that. I suggest the authors delete this sentence.

The reason why we include these studies is to show that we have familiarized ourselves with previous studies that applied the same modeling approach, thereby showcasing that the model is well established in research and practice. In fact, INCLUDE/CONCLUDE is so far the only raster-based modeling approach that has been applied at continental scales to model internal migration patterns due to SLR and other climate change impacts available in the current literature, and therefore constitutes the current state-of-the-art in raster-based migration modeling.

15. One line 291, the authors write about "attractiveness" But they don't use economic attractiveness. This seems like a major exclusion. Can the authors justify this exclusion?

Gravity models are based on the underlying assumption that densely populated locations are more attractive for human settlement than less densely populated locations due to the aggregate pull of multiple factors, which include economic factors such as job opportunities and income differentials (Anderson 2011). Furthermore, this attractiveness decreases with increasing distance due to factors such as transport costs and travel times (Rich 1980). We have added text to lines 352-355 for clarification.

16. On line 296, the authors write about A_i . What goes into A_i is buried in the Supplementary material. I suggest the authors include it here.

A_i is a calibrated local attractiveness factor established per country and raster cell during model calibration by eliminating the error produced when using the distance decay function only (see Reimann et al 2021 for further details). Following Reimann et al., we keep this factor constant based on the simplified assumption that local (un)attractiveness will persist until 2100. Establishing relationships between A_i and local conditions, for example using regression analysis, was beyond the scope of this study. We provide further context in lines 364-365.

17. On line 309, the authors write about the 2 degree by 2 degree grid from Kopp et al. 2 degrees is 7200 arc seconds, right? This, what, a 2000 meter resolution? But the MERIT DEM is 3 arc seconds and 90 m resolution. The ICZM is applied at 3, 4, and 6 arc seconds or 150 meters. The GLWD is 30 arc seconds or 900 meters. I don't think the GHS-POP resolution was mentioned in the paper. Because the spatial resolutions of all the various input data is so vastly different, I'm not sure exactly

what to conclude. I'd like to hear a justification from the authors about why this isn't an ecological fallacy.

We agree with the reviewer that we did not clearly describe the spatial resolution at which we processed the model input data. The SLR projections by Kopp et al. (Kopp et al 2017) were provided in point coordinates, which we merged to the nearest coastal segment (represented as a polyline shapefile) of the Mediterranean Coastal Database (MCD), as described in (Reimann et al 2018b). All datasets used to account for SLR and different adaptation strategies were harmonized to and processed at the spatial resolution of the MERIT DEM data (i.e. 3 arc seconds) before aggregating them to 30 arc seconds, the spatial resolution of CONCLUDE. In the aggregation process, we retained decimal values in hybrid raster cells, i.e. those cells partially located in the submerged area/retreat zone/setback zone. The population data were downloaded and processed at 30 arc seconds resolution. We have added these additional details in lines 369-370; 381-382; 408-412.

18. On line 310, the authors write the SLR amounts. These are only the 50th percentile values. What about the 80th percentile values? I went ahead and looked at the citations for these and the uncertainty intervals are provided in both citations (Table 1 in both). Why isn't the uncertainty in SLR propagated through your estimates? Furthermore, how in the world does SSP1 produce the most migrants when it has the least SLR?

We are aware that Kopp et al. provide the percentile ranges of the sea-level rise projections. However, as the focus of this study was on exploring the effects of different adaptation policies on SLR-related migration, we refrained from adding further complexity based on the insight that the effect of SLR on migration patterns was limited (please see also our responses to comments 7 and 10).

19. On line 345, the authors write that they "extended with the population data of GHS-POP." I'm unfamiliar with these datasets so I'm unsure exactly how the authors "extended the population data of the GHS-POP". Perhaps the authors could explain how they did this.

We added the GHS-POP data to the Mediterranean Coastal Database (MCD) based on the process described in Wolff et al. (2018) (who followed (Vafeidis et al 2008)): we used the rasters of GHS-POP (in 30 arc seconds resolution) and summed up the number of people per coastal segment and elevation increment (based on MERIT DEM) using zonal map statistics functions. We have added further details to the text (lines 422-426).

20. The sentence on line 347 (starting with "Protected segments were established..."), This whole section needs to be spelled out. How did you establish a safety function? How did wealth increase in these areas? Did you model wealth? (I get how pop density might since you model population). This is entirely too ambiguous.

Many thanks for the comment. Due to a lack of empirical data on actual protection levels around the world (see footnote on page 13), we used modelled data on the implementation of hard protection measures from the Dynamic Interactive Vulnerability Assessment (DIVA) modeling framework. Following Hinkel et al. (2014), protection measures are implemented following a demand for safety function derived as follows:

$$\frac{B}{Y} = \alpha(1+S)^{\theta} y^{\lambda} P^{\epsilon} F^{\theta} \quad 0 \leq \theta \leq 1,$$

where B is the benefit of coastal protection assuming a specific design return period (F), Y is GDP, y is per capita income, S is sea-level rise and P is population density. For further details please see Hinkel et al. (2014). However, as modeling the spatial feedbacks between SLR, protection, and spatial population distributions, and feeding the updated population rasters into DIVA in each time step was beyond the scope of this study, we used the simplified assumption that protection measures would be built in 2020 and raised with SLR, with no additional protection measures built until 2100. To clarify this point, we have added text in lines 426-433.

21. The sentence on line 350 starting with "As the number and locations of protected segments based on the DIVA results were similar across scenarios...", This is vaguely written and I'm unsure what's happening. DIVA basically said the scenarios don't matter so you post-processed them? How did you post-process them? This is entirely too ambiguous.

We agree with the reviewer that the rationale behind post-processing the DIVA results was not described clearly. We have reworked the entire section for clarification (lines 434-440).

22. On line 355, the authors write that they "excluded all rural segments..." Why? Justify this assumption.

We excluded all rural segments in 'Build with Nature' and 'Save Yourself' following the qualitative assumptions of the coastal SPA narratives (see Supplementary Text 1 and Supplementary Table 1) that only locations with high population densities would be protected. We clarify this point in lines 440-443.

23. On line 361, the authors write about the km of protected coastline. Are there cost estimates for armoring that are per km of coastline to get a sense of how reasonable these assumptions are? I have no idea if 14,000 km of protected coastline is even feasible. What if it costs like \$1 trillion?

3-27% is seems unreasonably wide, but, again, I have no real way of determining if this is appropriate based on the text as its currently written.

As the aim of this study was to explore the uncertainty space with regard to adaptation policies, involving different plausible protection levels under the respective scenario combination, we did not assess feasibility of these strategies in terms of costs. However, the protection assumptions are consistent with our coastal adaptation SPAs concerning the availability of adaptation funds and technological barriers to adaptation (Supplementary Table 1). We agree that the protected share of the coastline differs substantially across scenarios, but using this wide range allows us to explore the outcomes of the different adaptation policy scenarios in a meaningful way. Furthermore, these numbers are roughly in line with previous work that used cost-benefit analysis to establish stretches of the coastline where protection would be economically robust, resulting in 3.4 % (Lincke and Hinkel 2021) to 13.4 % (Lincke and Hinkel 2018) of the global coastline worth protecting, with higher shares along the Mediterranean coastline (see Figure 1 in Lincke and Hinkel (2018)). Considering the concentration of population and assets in the immediate Mediterranean coastal zone (Lange et al 2020), we deem a share of 27 % of coastline to be protected under the 'Hold the Line' scenario plausible. We added further justification to lines 453-459 and also added a map with the protection locations under each integrated scenario to the Supplementary Information to visually support our assumptions (Supplementary Figure 6).

24. Line 367, The "Accommodation" section. I'm not sure I would call any of this accommodation. Setback zones and coastal restoration is usually referred to as a "soft armoring". Accommodation might be deploying pumps, or changing to more saline tolerant crops. Those accommodation strategies are absent in this analysis. I'd suggest either dropping this part and rerunning your analysis or actually including some real accommodation strategies.

Please see our response to comment 5 (3) for the rationale behind accounting for setback zones as the only accommodation strategy in this study.

25. On line 372, the authors write that the ICZM has a distance of 100m but then the authors use a buffer of 6 arc seconds or 150 m. 150m >100m. Why choose 6 arc seconds? Why choose a buffer that's 50% larger than the statutory buffer?

Due to the spatial resolution at which the data were processed (i.e. 3 arc seconds), we had to approximate the coastline buffer, assuming that it would be extended under the sustainability-focused 'Build with Nature' scenario; would remain roughly the same under the 'Hold the Line' scenario; and would not be implemented under 'Save Yourself'. Please see lines 469-475 for clarification.

26. On line 378, the authors write about the "we assumed additional setback zones in areas at risk from regular coastal flooding to facilitate the restoration of coastal wetlands." Is the entire Mediterranean coastal wetlands? I'm unfamiliar with the local geography of the Med region. I don't know if this is a good assumption. And the authors aren't helping me along. Presumably, the entire coastline isn't coastal wetlands but will some areas that _aren't_ wetlands be included in this buffer?

We use two wetlands datasets to establish locations of wetlands along the Mediterranean coastline as described in lines 486-495, which we combine with the modeled floodplain in those locations not protected by hard protection measures to establish those coastal wetlands available for restoration, following the assumptions of Schuerch et al. (2018). We have reworked the entire section – also in response to comments 27 and 28 – for clarification (lines 476-500). Please see also Supplementary Figure 6 for the setback zones as implemented across the entire Mediterranean region (purple colors).

27. On line 380, the authors write about adding "the ESL height to the respective amount of SLR for each time step." Why is this mentioned here but not in the modelling SLR section? Where does this come from? The Kopp citation?

We describe the same approach (i.e. the bathtub approach) in the section 'Modeling SLR-induced migration'; however, as we additionally account for extreme sea levels for the accommodation (also retreat) assumptions, we include this explanation here as well. We rephrased the sentence (lines 478-481).

28. On line 383, the authors describe the 25- and 50-year Return Periods as "reflecting the differences in adaptive capacity between these regions." But I fail to see how how extreme sea levels or even return levels associated with flooding are related to adaptive capacity? Perhaps the authors can explain this?

We use these return periods to determine the extent of the setback zones, which we assume to be larger in the North than the South and East due to the higher adaptive capacity in the North. We have

rephrased the sentence for clarification (lines 478-485).

29. On line 394, the authors write that they make some areas "no longer available for human settlement, without removing population from these cells." Why? I think this is the only assumption that purposefully keeps people in place? What's the justification?

We realized that this sentence was misleading and rephrased it accordingly to "We then integrated the setback zones in CONCLUDE by classifying all land that fell into these zones per time step as no longer available for human settlement, thereby inhibiting migration towards these zones" (lines 497-500).

30. On line 398, the authors introduce the use of a 2-year return period in the southern and eastern Mediterranean and the 5-year return period in the northern Mediterranean. But why? There is (1) no justification provided for why they chose these specific return periods and (2) why they are different for each region. Reading between the lines with charity, it's possible the authors are alluding to the "differences in adaptive capacity between these regions" on line 383. But again, I can't see how return periods are related to adaptive capacity.

Indeed, we selected different return periods due to the differences in adaptive capacity across the two geographical regions for both, the setbacks and managed retreat zones. We inspected different return periods visually to ensure that the respective zones have a plausible size. Previous work used the 10-year return period for establishing managed retreat zones (Lincke and Hinkel 2021), which we found too large for this application. To clarify these points, we have added further context in lines 481-485 and 505-510.

31. Regarding the code availability statement. This is completely unacceptable in 2022. Please provide the code to understand what you are doing. Especially since the assumptions are poorly laid out. "available upon request" basically signifies the code is not available in practice.

We agree with the reviewer and have deposited the model code in a GitHub repository (see code availability statement).

Reviewer #2 (Remarks to the Author):

An interesting paper adding to the canon of work on possible migration futures occurring in response to sea level rise. I found the methodology robust within the demography paradigm by using a spatially-explicit gravity-based model. If I had a criticism of this paper it would be around whether the assumptions of gravity-based models will hold into the future, for example, whether 'city islands' ringed by sea defences will continue to act as attractors and whether the bigger motivation for migration is more likely to be repeated storm surge events and decreasing returning displaced populations (e.g. see Fussell, E., Sastry, N. and Van Landingham, M., 2010. Race, socioeconomic status, and return migration to New Orleans after Hurricane Katrina. *Population and environment*, 31(1), pp.20-42). Thus, to my reading the paper neglects to question assumptions around the way in which people are envisaged to respond to climate stresses. This type of model is useful for modelling slow demographic change but has some important limitations in modelling migration flows and the multiple and novel ways climate change might affect migration. So in summary a solid piece of work that maybe could be more reflexive over the assumptions of the model.

We would like to thank Reviewer #2 for the time and effort taken to review our manuscript and for their positive feedback on our work. We agree with the thought-provoking points raised and reflect upon them in the 'Insights for policy making' and 'Needs for future research' sections (lines 235-251 and 276-285). We believe that the changes made in the manuscript raise important points for future research on sea-level rise migration and the broader climate migration context.

Reviewer #3 (Remarks to the Author):

This paper presents an interesting set of findings examining a calibration of a gravity model called CONCLUDE to the Mediterranean region, coupled with a model of SLR out to 2100 and a set of adaptation scenarios, to examine the effects of hard adaptation measures on coastal outmigration, differentiated between the comparatively rich North Mediterranean and the comparatively poor South-East Mediterranean.

The paper is clear and well structured. I will not provide a list of specific criticisms. I outline only two potential flaws that might stand in the way of it being a valid and important contribution. Both issues relate to the reliance on scenarios as the key experiment/shock.

We would like to thank Reviewer #3 for their efforts reviewing our manuscript. They raised two important aspects that needed clarification: we have now revised the manuscript to clarify the assumptions taken from the SSPs (including international migration) as well as the type of feedbacks (i.e. spatial feedbacks) accounted for in this study. We believe that we have addressed the points raised in a satisfactory manner. Please see our responses to the comments below.

First, migration is already implicit within the SSPs. Your methods don't discuss this, and leave out the migration element when describing the SSPs applied to your model. To be valid, you need to demonstrate how allowing migration to be endogenous to your model – and possibly heterogeneous across place – is not inconsistent with the SSPs. In particular, you should demonstrate that the migration flows that emerge in your simulation do not violate their assumed states in the SSPs (or importantly, violate them differently across your rich and poor regions). (Doing this well, in my mind, makes it an even more important contribution, as a roadmap for future researchers to see better how to do this – draw on the SSPs as backbones for modeled processes that are at least partially endogenous to them).

We agree that the assumptions taken from the SSPs (i.e. endogenous) versus those developed for this study (i.e. exogenous) were not clearly stated in the previous version of the manuscript. As we model internal migration only, our work is fully consistent with the population and urbanization projections provided on the IIASA SSP database (IIASA 2018); these include assumptions on fertility, mortality, and international migration as well as urbanization rates and levels, respectively. Furthermore, our model assumptions on spatial development patterns and adaptation policies (i.e. SPAs) are consistent with the qualitative narratives of the SSPs. We have now reworked the entire manuscript to clarify that we model internal migration only and that our work is fully consistent with the SSPs (lines 89-101; 369-372).

Second, it is a mis-statement to say that your contribution is the first to capture feedbacks between among population, adaptation policy and SLR, because only population is endogenous to your model. SLR and adaptation policy are fixed by your scenarios, and are thus providing bounding conditions for population to evolve, nothing more. The only mechanism through which feedbacks can happen in your model are the basic premises of the gravity model (bigger cities pull more) which are more than a century old and are not your contribution. If adaptations were an endogenous response to population change (as you might get from an ABM), this could possibly be part of a feedback, but to my understanding, this is not what you have modeled. Thus, the feedback language needs to be removed and the contribution clarified, or a clear demonstration of the feedback (i.e., showing in a

causal loop diagram a set of endogenous processes in your model that complete a feedback loop and include the novel adaptation and SLR features of your model) needs to be provided.

We would like to thank the reviewer for raising this point. We agree that we do not account for dynamic feedbacks between SLR and emergent behavior (i.e. adaptation decisions) as could be done with the help of an agent-based modeling approach. We reflect upon this issue as one future research need (lines 270-279).

In the context of this study, we are referring to spatial feedbacks where spatial changes in population distributions reinforce observed population patterns, particularly driven by the implementation of hard protection measures. This effect is referred to as the levee effect or safe development paradox (e.g. Di Baldassarre et al 2015, 2018, Mård et al 2018). We have revised the entire manuscript to ensure that we clearly define the type of feedbacks we are referring to and consistently use the term 'spatial feedback' throughout the manuscript.

Furthermore, we have changed the manuscript title accordingly, which now reads "Beyond migrant numbers – exploring spatial feedbacks between adaptation policies and internal migration patterns due to sea-level rise".

I look forward to a version of this manuscript that addresses the above issues, as I believe it will be quite a well-read contribution to the literature.

We hope that the revised manuscript fulfills the expectations of the reviewer.

References

- Anderson J E 2011 The Gravity Model *Annu. Rev. Econom.* **3** 133–60
- Di Baldassarre G, Kreibich H, Vorogushyn S, Aerts J, Arnbjerg-Nielsen K, Barendrecht M, Bates P, Borga M, Botzen W, Bubeck P, De Marchi B, Llasat C, Mazzoleni M, Molinari D, Mondino E, Mård J, Petrucci O, Scolobig A, Viglione A and Ward P J 2018 Hess opinions: An interdisciplinary research agenda to explore the unintended consequences of structural flood protection *Hydrol. Earth Syst. Sci.* **22** 5629–37
- Di Baldassarre G, Viglione A, Carr G, Kuil L, Yan K, Brandimarte L and Blöschl G 2015 Debates- Perspectives on socio-hydrology: Capturing feedbacks between physical and social processes *Water Resour. Res.* **51** 4770–81
- Bell A R, Wrathall D J, Mueller V, Chen J, Oppenheimer M, Hauer M, Adams H, Kulp S, Clark P U, Fussell E, Magliocca N, Xiao T, Gilmore E A, Abel K, Call M and Slangen A B A 2021 Migration towards Bangladesh coastlines projected to increase with sea-level rise through 2100 *Environ. Res. Lett.* **16** 024045
- Hinkel J, Lincke D, Vafeidis A T, Perrette M, Nicholls R J, Tol R S J, Marzeion B, Fettweis X, Ionescu C and Levermann A 2014 Coastal flood damage and adaptation costs under 21st century sea-level rise *Proc. Natl. Acad. Sci. U. S. A.* **111** 3292–7
- IIASA 2018 SSP Database *Version 2.0* Online: <https://tntcat.iiasa.ac.at/SspDb>
- IPCC 2022 *Climate Change 2022: Impacts, Adaptation, and Vulnerability. Contribution of Working Group II to the Sixth Assessment Report of the Intergovernmental Panel on Climate Change* ed H-O Pörtner, D C Roberts, M Tignor, E S Poloczanska, K Mintenbeck, A Alegría, M Craig, S Langsdorf, S Löschke, V Möller, A Okem and B Rama (Cambridge University Press. In Press)
- Kopp R E, DeConto R M, Bader D A, Hay C C, Horton R M, Kulp S, Oppenheimer M, Pollard D and Strauss B H 2017 Evolving Understanding of Antarctic Ice-Sheet Physics and Ambiguity in Probabilistic Sea-Level Projections *Earth's Futur.* **5** 1217–33
- Lange M A, Llasat M C, Snoussi M, Graves A, Le Tellier J, Queralt A and Vagliasindi G M 2020 Introduction *Climate and Environmental Change in the Mediterranean Basin – Current Situation and Risks for the Future. First Mediterranean Assessment Report* ed W Cramer, J Guiot and K Marini (Marseille, France: Plan Bleu, UNEP/MAP) pp 41–58
- Lincke D and Hinkel J 2021 Coastal Migration due to 21st Century Sea-Level Rise *Earth's Futur.* **9** 1–14
- Lincke D and Hinkel J 2018 Economically robust protection against 21st century sea-level rise *Glob. Environ. Chang.* **51** 67–73
- Lincke D, Wolff C, Hinkel J, Vafeidis A, Blickensdörfer L and Povh Skugor D 2020 The effectiveness of setback zones for adapting to sea-level rise in Croatia *Reg. Environ. Chang.* **20** 46
- Mård J, Di Baldassarre G and Mazzoleni M 2018 Nighttime light data reveal how flood protection shapes human proximity to rivers *Sci. Adv.* **4** 1–8
- Merkens J-L, Reimann L, Hinkel J and Vafeidis A T 2016 Gridded population projections for the coastal zone under the Shared Socioeconomic Pathways *Glob. Planet. Change* **145** 57–66
- de Moel H, Aerts J C J H and Koomen E 2011 Development of flood exposure in the Netherlands during the 20th and 21st century *Glob. Environ. Chang.* **21** 620–7
- Neumann B, Vafeidis A T, Zimmermann J and Nicholls R J 2015 Future coastal population growth and exposure to sea-level rise and coastal flooding--a global assessment *PLoS One* **10** e0118571

- Nicholls R J 2004 Coastal flooding and wetland loss in the 21st century *Glob. Environ. Chang.* **14** 69–86
- Oppenheimer M, Glavovic B C, Hinkel J, van de Wal R, Magnan A K, Abd-Elgawad A, Cai R, Cifuentes-Jara M, DeConto R M, Ghosh T, Hay J, Isla F, Marzeion B, Meyssignac B and Sebesvari Z 2022 Sea Level Rise and Implications for Low-Lying Islands, Coasts and Communities *The Ocean and Cryosphere in a Changing Climate* ed H-O Pörtner, D-C Roberts, V Masson-Delmotte, P Zhai, M Tignor, E Poloczanska, K Mintenbeck, A Alegría, M Nicolai, A Okem, J Petzold, B Rama and N M Weyer (Cambridge University Press) pp 321–446
- Reimann L, Jones B, Nikolettopoulos T and Vafeidis A T 2021 Accounting for internal migration in spatial population projections—a gravity-based modeling approach using the Shared Socioeconomic Pathways *Environ. Res. Lett.* **16** 074025
- Reimann L, Merkens J-L and Vafeidis A T 2018a Regionalized Shared Socioeconomic Pathways: narratives and spatial population projections for the Mediterranean coastal zone *Reg. Environ. Chang.* **18** 235–45
- Reimann L, Vafeidis A T, Brown S, Hinkel J and Tol R S J 2018b Mediterranean UNESCO World Heritage at risk from coastal flooding and erosion due to sea-level rise *Nat. Commun.* **9** 4161
- Rich D C 1980 *Potential models in human geography* vol 26 (Norwich: Geo Abstracts)
- Schuerch M, Spencer T, Temmerman S, Kirwan M L, Wolff C, Lincke D, McOwen C J, Pickering M D, Reef R, Vafeidis A T, Hinkel J, Nicholls R J and Brown S 2018 Future response of global coastal wetlands to sea-level rise *Nature* **561** 231–4
- Vafeidis A T, Nicholls R J, McFadden L, Tol R S J, Hinkel J, Spencer T, Grashoff P S, Boot G and Klein R J T 2008 A New Global Coastal Database for Impact and Vulnerability Analysis to Sea-Level Rise *J. Coast. Res.* **244** 917–24
- van Vuuren D P, Kriegler E, O'Neill B C, Ebi K L, Riahi K, Carter T R, Edmonds J, Hallegatte S, Kram T, Mathur R and Winkler H 2014 A new scenario framework for Climate Change Research: scenario matrix architecture *Clim. Change* **122** 373–86
- Wolff C, Vafeidis A T, Muis S, Lincke D, Satta A, Lionello P, Jimenez J A, Conte D and Hinkel J 2018 A Mediterranean coastal database for assessing the impacts of sea-level rise and associated hazards *Sci. Data* **5** 180044

REVIEWER COMMENTS

Reviewer #1 (Remarks to the Author):

I want to thank the authors for their thoughtful feedback and work on their manuscript. The clarifications to the figures, text, methods, and supplementary material have greatly improved the manuscript.

But there are several comments that the authors did not respond to. I'm not sure if these issues preclude publication of the paper, but if I were to read the published version of this paper (as someone who did not review it), I would raise the same questions. Specifically, the results appear implausible and the authors have not adequately responded to my comments about their implausibility. The authors wish to have a results section that spells out the exact timing and numbers of migrants (Figure 2 and its associated text) but do not wish to incorporate the uncertainty inherent in these projections -- choosing to *only* model the median values of each scenario (SSP, SLR, etc.). This produces results with a false sense of precision. On one hand, we have exact results and timings but the authors want us to only view their results as if it only explores "the effects of different adaptation policies on SLR-related migration" (Authors response to comment 18) rather than on the timings and numbers of migrants (see also "However, we would like to emphasize that we do not aim to predict the number of migrants in the course of the 21st century, but that we rather explore the plausible range of migrant numbers if such adaptation policies were pursued." authors response to comment 7). I do not object to framing their results as predictive (exact numbers/timings with appropriate uncertainty intervals) or to framing their results as general explorations (exploring how different adaptation policies impacts migration) but I *do* object to producing predictive results without appropriate uncertainty that are framed as exploratory.

Furthermore, the believability of any modeling endeavor relies on how well the model mimics reality. Why would I believe any model that cannot reflect reality? The authors' response to comment 7 is thus woefully inadequate. 3 million migrants during this decade is likely implausible and will not reflect reality. We are already 20% (2 years) into the decade and have there been 600K migrants? The authors did not comment on the feasibility of their results and instead want to frame their results as an exploration. But if their model cannot reflect reality in the short term -- where prediction uncertainty is generally low, how can we believe their model in the long term -- when prediction uncertainty is generally high)? Approximately one-quarter of all SLR-related migration will occur this decade? This is simply ridiculous and suggests there is a modeling or assumption error buried deep within their analysis.

The authors response to comment 7 regarding the cascade effect is also woefully inadequate. The "acceleration of SLR in the second half of 21st century" is not a sudden onset event, as their results in Figure 2 suggest. It is a gradual, smooth acceleration. I understand that each SLR scenario implies varying curves and timings but not such to create such "bumps" in the amount of SLR. This, too, suggests an implausibility in the modeling that suggests there is some modeling or assumption error buried deep within the analysis.

Regarding the authors response to comment 9, again, how does SLR cause Paris and Madrid to LOSE population? This is very important but the authors did not respond nor are there is their any text in the manuscript addressing this. Related, the response to Comment 11 is still inadequate. The zoomed in figure 4 still shows that adaptation policies lead to out-migration in the Nile and Nile delta regions but Supp Figure 5 shows moderate to high confidence for *in-migration* to this region.

I do not believe, and the authors have not convinced me, that the Mediterranean region will see 3 million SLR migrants this decade, that statistical uncertainty needs not be modelled, nor how SLR migration will cause population losses in major inland cities. There are also differences between the published figures and the supplementary figures that remain unresolved.

Reviewer #2 (Remarks to the Author):

I felt the authors have done well to address the reviewers comments and have a publishable paper. My only slight concern was about giving a projection from 2020 when by we approaching the end of 2022 but presume this represents the gap between research and publication.

Reviewer #3 (Remarks to the Author):

I appreciate the effort the authors have taken to respond to the concerns of the 3 reviewers. The revised version does a great job of situating the manuscript in the current literature on modeling coastal adaptation.

Reviewer #1 (Remarks to the Author):

I want to thank the authors for their thoughtful feedback and work on their manuscript. The clarifications to the figures, text, methods, and supplementary material have greatly improved the manuscript.

We would like to thank the reviewer for their time and effort taken to review the manuscript a second time. Based on the remaining comments, we realized that we did not address some of the points raised in the first round of revisions in sufficient detail. Therefore, we have revised the entire manuscript carefully and added a new section "Be aware of uncertainties" that contextualizes the uncertainties inherent in the study, supported by additional literature. Please see our point-by-point responses to the remaining issues below. All changes made to the manuscript are marked as track changes. The references added to the list of references are highlighted in yellow. We sincerely hope that the manuscript is now acceptable for publication.

1. But there are several comments that the authors did not respond to. I'm not sure if these issues preclude publication of the paper, but if I were to read the published version of this paper (as someone who did not review it), I would raise the same questions. Specifically, the results appear implausible and the authors have not adequately responded to my comments about their implausibility. The authors wish to have a results section that spells out the exact timing and numbers of migrants (Figure 2 and its associated text) but do not wish to incorporate the uncertainty inherent in these projections -- choosing to **only** model the median values of each scenario (SSP, SLR, etc.). This produces results with a false sense of precision. On one hand, we have exact results and timings but the authors want us to only view their results as if it only explores "the effects of different adaptation policies on SLR-related migration" (Authors response to comment 18) rather than on the timings and numbers of migrants (see also "However, we would like to emphasize that we do not aim to predict the number of migrants in the course of the 21st century, but that we rather explore the plausible range of migrant numbers if such adaptation policies were pursued." authors response to comment 7). I do not object to framing their results as predictive (exact numbers/timings with appropriate uncertainty intervals) or to framing their results as general explorations (exploring how different adaptation policies impacts migration) but I **do** object to producing predictive results without appropriate uncertainty that are framed as exploratory.

As pointed out previously, this is an exploratory study that aims to assess ranges of migrant numbers under different adaptation policy scenarios (we added text for further emphasis in lines 96-99 and 186-189), and the numbers provided should not be interpreted as predictions. In order to address the issues raised by the reviewer, we have made the following considerable changes to the manuscript:

- **Using ranges in results:** *we have replaced exact numbers (which seem to give a false sense of accuracy) with ranges or approximate values throughout the results section 'The effects of adaptation on total migrant numbers' (lines 102-154; caption Fig. 2).*
- **Emphasize indicative trends:** *we emphasize that the numbers presented here reflect indicative trends of future internal migrants and should be interpreted with caution due to a range of uncertainties (lines 96-99; caption Fig. 2).*

- **New section on uncertainty:** we have added reflections on these uncertainties in a new section ‘Be aware of uncertainties’, particularly concentrating on the scenario assumptions, migration modeling approach, and input data used (lines 247-311). However, extensively reporting on and systematically quantifying the uncertainties related to scenarios and projections (e.g. SSPs, RCPs, different percentiles); input data (e.g. population, elevation, settlements, coastal wetlands); and migration modeling approaches (e.g. gravity-based modeling, agent-based modeling) goes beyond the scope of this work (as pointed out in lines 306-309). We therefore propose conducting a systematic sensitivity analysis in follow-up research (lines 344-347).
- **Scenario-based approach:** while we are well aware that different percentiles exist for the sea-level rise (SLR) projections used in this study, we intentionally use the median values (following Reimann et al., 2018) as we account for uncertainty in future SLR by using three SLR scenarios based on RCPs 2.6, 4.5, and 8.5, which is common practice in scenario-based modeling studies (e.g. Bachner et al., 2022; Kirezci et al., 2023; Lincke et al., 2020; Reimann et al., 2018; Rohat et al., 2020; Schuerch et al., 2018; Tiggeloven et al., 2020; Vafeidis et al., 2019). Compared to the medians reported in the IPCC’s 6th Assessment Report (AR6), the SLR projections used here are on the low end of the uncertainty range (Ali & Cramer, 2022; Fox-Kemper et al., 2021) and more in line with those values reported in the Special Report on the Ocean and Cryosphere in a Changing Climate (SROCC) (Oppenheimer et al., 2022). Using these SLR projections, we anticipate our results to be rather conservative estimates of the number of internal migrants due to SLR than overestimations (for the no adaptation policy scenarios). We have added text in 448-454 to reflect on this point.

With these revisions, we are convinced that we sufficiently represent the future uncertainty space, particularly bearing in mind that the focus of this study is on the exploration of the potential effect of coastal adaptation policy scenarios on SLR-related internal migration rather than the effect of SLR alone. Related to this point and stated in lines 103-106 and 191-194 (see also Fig. 2a), the effect of the amount of SLR on the number of migrants is small compared to the effect of the adaptation policy, with about 20 million internal migrants until 2100 independent from the SLR scenario used; only the timing of migration differs across SLR scenarios, with a later increase in the number of migrants under the lowest SLR scenario (i.e. RCP2.6).

2. Furthermore, the believability of any modeling endeavor relies on how well the model mimics reality. Why would I believe any model that cannot reflect reality? The authors’ response to comment 7 is thus woefully inadequate. 3 million migrants during this decade is likely implausible and will not reflect reality. We are already 20% (2 years) into the decade and have there been 600K migrants? The authors did not comment on the feasibility of their results and instead want to frame their results as an exploration. But if their model cannot reflect reality in the short term -- where prediction uncertainty is generally low, how can we believe their model in the long term -- when prediction uncertainty is generally high)? Approximately one-quarter of all SLR-related migration will occur this decade? This is simply ridiculous and suggests there is a modeling or assumption error buried deep within their analysis.

We of course agree with the reviewer that the modeling approach should be able to replicate observed migration patterns. Indeed, the CONCLUDE model used in this study has been extensively calibrated and validated in previous work, replicating past internal migration patterns well, with an R^2 of 0.97 and a root mean squared error (RMSE) of 79.4 people (Reimann et al., 2021). We must note,

however, that when projecting future internal migration patterns, accounting for the potential effects of SLR and adaptation policies on migration, validation is particularly challenging as empirical data of observed impacts of SLR and adaptation policies on internal migration do not exist, particularly not at this scale of analysis (see lines 282-290). To tackle this limitation, local surveys can be conducted which can provide the basis for upscaling the insights to the continental scale; however, such survey are rare and require long time frames (Duijndam et al., 2022). We discuss this need for future work in lines 355-358.

We would like to emphasize that in our study we model internal migration (as opposed to international migration). As this was not clear enough in our previous version, we have revised the entire manuscript to point this out clearly. Specifically, independent from the implementation of adaptation policies, gradual submergence of low-lying coastal land due to SLR will force people in affected locations to migrate from their place of residence (see lines 463-472 in the Methods) – these migration processes can take place over short distances (i.e. the next raster cell) and the same people may be forced to migrate several times in the course of the century (see text added to lines 189-191 and 468-469 for clarification). A similar effect can be observed in the ‘Build with Nature’ adaptation policy scenario, where retreat and setback zones are expanded with increasing SLR, resulting in a gradual increase in the number of people that are forced to migrate from unprotected coastal stretches (see lines 566-571; 579-581). This effect of multiple movements is captured in CONCLUDE and leads to the high numbers of migrants that the reviewer has mentioned, which refer to numbers of individual movements rather than to people migrating.

As a first-order validation exercise, we compared our results for the first two years of the modeling period (as suggested by the reviewer) to the number of internally displaced people due to disasters caused by a range of hazards as reported in the Global Internal Displacement Database (GIDD) (IDMC, 2022). We have found that the number of internal migrants projected in this study (i.e. roughly 600,000) has a similar magnitude to the number of internal displacements observed in the Mediterranean region in 2020 and 2021, which amounted to more than 450,000 internally displaced people. Of course, these numbers are hardly comparable due to different hazards and migration types captured, but this comparison shows that we project a reasonable magnitude of internal migrants.

3. The authors response to comment 7 regarding the cascade effect is also woefully inadequate. The "acceleration of SLR in the second half of 21st century" is not a sudden onset event, as their results in Figure 2 suggest. It is a gradual, smooth acceleration. I understand that each SLR scenario implies varying curves and timings but not such to create such "bumps" in the amount of SLR. This, too, suggests an implausibility in the modeling that suggests there is some modeling or assumption error buried deep within the analysis.

Indeed, the effect of SLR as implemented in CONCLUDE is gradual. However, as pointed out previously, “tipping points” occur when densely populated raster cells are submerged due to SLR, leading to accelerated forced (autonomous) migration of the population residing in these cells. After carefully double-checking the data (i.e. flooded raster cells) produced as model inputs for each SLR scenario and decadal time step to represent those zones submerged by SLR, we found that these “tipping points” result from a few raster cells that experience submergence from one time step to the next located around the city of Alexandria, which is home to more than 5 million inhabitants (United Nations Population Division, 2019). The figure below illustrates this effect for each SSP-RCP combination (no adaptation policies; see also Figure CCP4.6 in IPCC AR6 (p. 2244) (Ali & Cramer, 2022) for illustration of the same effect). We found that these migration tipping points are dominated

by the number of migrants projected for Egypt, which make up 82 % (SSP5-RCP8.5) to 90 % (SSP1-RCP2.6) of all internal migration projected for the respective time steps (i.e. 2080/90 in SSP1-RCP2.6; 2060/70 in SSP3-RCP4.5 and SSP5-RCP8.5). We added text to lines 124-126 to point out this bias. Furthermore, we added a Supplementary Data file that reports the total number of internal migrants by country, scenario combination, and time step.

The observed effect is potentially further amplified by the population data (i.e. GHS-POP) used as model input, which use built-up land derived from satellite imagery to model spatial population distributions (Florczyk et al., 2019). This spatial population modeling approach tends to over-concentrate the population as not all built-up land is detected in satellite imagery (Leyk et al., 2018; MacManus et al., 2021). Therefore, we may overestimate the number of migrants in those coastal locations, where the population is over-concentrated. In order to better understand the effects of different input datasets (such as population, urban definition, elevation) on the projected number and spatial patterns of migration, we would encourage future work to conduct a systematic sensitivity analysis (see also our response to comment 1). We have added reflections on the population data to lines 297-301.

“Tipping points” in submergence of low-lying coastal land around the city of Alexandria per scenario combination and time step, assuming no adaptation policies. a) SSP1-RCP2.6; b) SSP3-RCP4.5; c) SSP5-RCP8.5. Basic World Cities Database: SimpleMaps. Basic World Cities Database. <https://simplemaps.com/data/world-cities> (2022). - CC BY 4.0

4. Regarding the authors response to comment 9, again, how does SLR cause Paris and Madrid to LOSE population? This is very important but the authors did not respond nor are there is their any text in the manuscript addressing this. Related, the response to Comment 11 is still inadequate. The zoomed in figure 4 still shows that adaptation policies lead to out-migration in the Nile and Nile delta regions but Supp Figure 5 shows moderate to high confidence for *in-migration* to this region.

The reason for zooming into the maps in Figure 4 was to increase the readability of the figure. For the same reason, we changed the manuscript text as Madrid and Paris were no longer shown in the maps. We would like to emphasize that the figure presents spatial patterns of internal migration until 2100, a) without adaptation policies (upper row), b) with adaptation policies (middle row), and c) comparing the two (lower row). While rows a) and b) appear to result in very similar spatial migration patterns, row c) shows that the implementation of adaptation policies largely reverses spatial migration patterns, mainly due to the so-called 'levee effect' (please see lines 167-183 for further detail). In total, we see out-migration from the immediate coastal zone in the Nile delta under the 'no adaptation policies' and 'with adaptation policies' scenarios (rows a) and b)), with considerably fewer migrants when adaptation policies are implemented (as seen in c)).

The same effect can be observed in Paris and Madrid: In total numbers, both the lack of adaptation policies (Supplementary Figure 2) as well as the implementation of adaptation policies (Supplementary Figure 3) result in internal migration towards these cities. When comparing the effect of adaptation policies with the lack of adaptation policies on SLR-related internal migration (Supplementary Figure 4), we see that the implementation of coastal adaptation measures results in fewer people migrating to these cities compared to the no adaptation policies reference protections.

These patterns are confirmed in the migration hotspot analysis as presented in Fig. 5 and Supplementary Figure 5. We reworked Fig. 4 along with the figure caption for clarification.

I do not believe, and the authors have not convinced me, that the Mediterranean region will see 3 million SLR migrants this decade, that statistical uncertainty needs not be modelled, nor how SLR migration will cause population losses in major inland cities. There are also differences between the published figures and the supplementary figures that remain unresolved.

We believe that the changes made in the manuscript and our responses to the reviewer's comments above address the remaining points raised, presenting a convincing case of plausible future internal migration related to SLR in the Mediterranean region under a range of adaptation policy scenarios.

Reviewer #2 (Remarks to the Author):

I felt the authors have done well to address the reviewers comments and have a publishable paper. My only slight concern was about giving a projection from 2020 when by we approaching the end of 2022 but presume this represents the gap between research and publication.

We would like to thank Reviewer #2 for their positive feedback.

The reason for giving a projection from 2020 is an offset in the input data: we used the GHS-POP data of 2019, which provided population rasters for the years 1975, 1990, 2000, and 2015 based on

Landsat imagery. While a new version of GHS-POP has become available in late 2022 (including the year 2020), we refrained from updating the results of this study to ensure consistency with the ‘no SLR’ projections used as the baseline scenarios (see Reimann et al., 2021). We provide a brief explanation in a footnote on page 14.

Reviewer #3 (Remarks to the Author):

I appreciate the effort the authors have taken to respond to the concerns of the 3 reviewers. The revised version does a great job of situating the manuscript in the current literature on modeling coastal adaptation.

We would like to thank Reviewer #3 for their efforts reviewing our manuscript a second time, and thank the reviewer for their positive evaluation.

References

- Ali, E., & Cramer, W. (2022). CCP 4 Mediterranean Region. In *Climate Change 2022: Impacts, Adaptation and Vulnerability. Contribution of Working Group II to the Sixth Assessment Report of the Intergovernmental Panel on Climate Change*.
<https://doi.org/10.1017/9781009325844.021.2233>
- Bachner, G., Lincke, D., & Hinkel, J. (2022). The macroeconomic effects of adapting to high-end sea-level rise via protection and migration. *Nature Communications*, 13(1), 5705.
<https://doi.org/10.1038/s41467-022-33043-z>
- Duijndam, S. J., Botzen, W. J. W., Hagedoorn, L. C., & Aerts, J. C. J. H. (2022). Anticipating sea-level rise and human migration: A review of empirical evidence and avenues for future research. *WIREs Climate Change*, 13(1). <https://doi.org/10.1002/wcc.747>
- Florczyk, A. J., Corbane, C., Ehrlich, D., Freire, S., Kemper, T., Maffenini, L., Melchiorri, M., Pesaresi, M., Politis, P., Schiavina, M., Sabo, F., & Zanchetta, L. (2019). GHSL data package 2019. In *EUR* (Vol. 29788). Publications Office of the European Union.
<http://www.worldcat.org/oclc/1112369877>
- Fox-Kemper, B., Hewitt, H. T., Xiao, C., Aðalgeirsdóttir, G., Drijfhout, S. S., Edwards, T. L., Gолledge, N. R., Hemer, M., Kopp, R. E., Krinner, G., Mix, A., Notz, D., Nowicki, S., Nurhati, I. S., Ruiz, L., Sallée, J.-B., Slangen, A. B. A., & Yu, Y. (2021). Ocean, Cryosphere and Sea Level Change. In V. Masson-Delmotte, P. Zhai, A. Pirani, S. L. Connors, C. Péan, S. Berger, N. Caud, Y. Chen, L. Goldfarb, M. I. Gomis, M. Huang, K. Leitzell, E. Lonnoy, J. B. R. Matthews, T. K. Maycock, T. Waterfield, O. Yelekci, R. Yu, & B. Zhou (Eds.), *Climate Change 2021: The Physical Science Basis. Contribution of Working Group I to the Sixth Assessment Report of the Intergovernmental Panel on Climate Change* (pp. 1211–1362). Cambridge University Press.
<https://doi.org/10.1017/9781009157896.011>
- IDMC. (2022). *Global Internal Displacement Database*. <https://www.internal-displacement.org/database/displacement-data>

- Kirezci, E., Young, I. R., Ranasinghe, R., Lincke, D., & Hinkel, J. (2023). Global-scale analysis of socioeconomic impacts of coastal flooding over the 21st century. *Frontiers in Marine Science*, 9(January), 1–21. <https://doi.org/10.3389/fmars.2022.1024111>
- Leyk, S., Uhl, J. H., Balk, D., & Jones, B. (2018). Assessing the accuracy of multi-temporal built-up land layers across rural-urban trajectories in the United States. *Remote Sensing of Environment*, 204, 898–917. <https://doi.org/10.1016/j.rse.2017.08.035>
- Lincke, D., Wolff, C., Hinkel, J., Vafeidis, A., Blickensdörfer, L., & Povh Skugor, D. (2020). The effectiveness of setback zones for adapting to sea-level rise in Croatia. *Regional Environmental Change*, 20(2), 46. <https://doi.org/10.1007/s10113-020-01628-3>
- MacManus, K., Balk, D., Engin, H., McGranahan, G., & Inman, R. (2021). Estimating population and urban areas at risk of coastal hazards, 1990–2015: how data choices matter. *Earth System Science Data*, 13(12), 5747–5801. <https://doi.org/10.5194/essd-13-5747-2021>
- Oppenheimer, M., Glavovic, B. C., Hinkel, J., van de Wal, R., Magnan, A. K., Abd-Elgawad, A., Cai, R., Cifuentes-Jara, M., DeConto, R. M., Ghosh, T., Hay, J., Isla, F., Marzeion, B., Meyssignac, B., & Sebesvari, Z. (2022). Sea Level Rise and Implications for Low-Lying Islands, Coasts and Communities. In H.-O. Pörtner, D.-C. Roberts, V. Masson-Delmotte, P. Zhai, M. Tignor, E. Poloczanska, K. Mintenbeck, A. Alegría, M. Nicolai, A. Okem, J. Petzold, B. Rama, & N. M. Weyer (Eds.), *The Ocean and Cryosphere in a Changing Climate* (pp. 321–446). Cambridge University Press. <https://doi.org/10.1017/9781009157964.006>
- Reimann, L., Jones, B., Nikolettopoulos, T., & Vafeidis, A. T. (2021). Accounting for internal migration in spatial population projections—a gravity-based modeling approach using the Shared Socioeconomic Pathways. *Environmental Research Letters*, 16(7), 074025. <https://doi.org/10.1088/1748-9326/ac0b66>
- Reimann, L., Vafeidis, A. T., Brown, S., Hinkel, J., & Tol, R. S. J. (2018). Mediterranean UNESCO World Heritage at risk from coastal flooding and erosion due to sea-level rise. *Nature Communications*, 9(1), 4161. <https://doi.org/10.1038/s41467-018-06645-9>
- Rohat, G., Monaghan, A., Hayden, M. H., Ryan, S. J., Charrière, E., & Wilhelmi, O. (2020). Intersecting vulnerabilities: climatic and demographic contributions to future population exposure to Aedes-borne viruses in the United States. *Environmental Research Letters*, 15(8), 084046. <https://doi.org/10.1088/1748-9326/ab9141>
- Schuerch, M., Spencer, T., Temmerman, S., Kirwan, M. L., Wolff, C., Lincke, D., McOwen, C. J., Pickering, M. D., Reef, R., Vafeidis, A. T., Hinkel, J., Nicholls, R. J., & Brown, S. (2018). Future response of global coastal wetlands to sea-level rise. *Nature*, 561(7722), 231–234. <https://doi.org/10.1038/s41586-018-0476-5>
- Tiggeloven, T., de Moel, H., Winsemius, H. C., Eilander, D., Erkens, G., Gebremedhin, E., Diaz Loaiza, A., Kuzma, S., Luo, T., Iceland, C., Bouwman, A., van Huijstee, J., Ligtvoet, W., & Ward, P. J. (2020). Global-scale benefit–cost analysis of coastal flood adaptation to different flood risk drivers using structural measures. *Natural Hazards and Earth System Sciences*, 20(4), 1025–1044. <https://doi.org/10.5194/nhess-20-1025-2020>
- United Nations Population Division, D. of E. and S. A. (2019). *World urbanization prospects*. United Nations.

Vafeidis, A. T., Schuerch, M., Wolff, C., Spencer, T., Merkens, J. L., Hinkel, J., Lincke, D., Brown, S., & Nicholls, R. J. (2019). Water-level attenuation in global-scale assessments of exposure to coastal flooding. *Natural Hazards and Earth System Sciences*, *19*(5), 973–984.
<https://doi.org/10.5194/nhess-19-973-2019>